# Treatment Responder Classification with Abstention

## Abstract

Treatment responder classification seeks to learn a rule to classify individuals who will benefit from the treatment. This paper studies a new scenario in treatment responder classification when abstention is allowed, i.e., practitioners can opt out of making uncertain classification on some individuals for further investigation. By revealing the implicit relation between causal misclassification risk with abstention and Conditional Value at Risk (CVaR), we develop a doubly robust method named **TRECA** to learn the classification rule under loose convergence conditions on nuisance parameters, and further extend it to deal with possible violation on key assumptions such as monotonicity and unconfoundedness. Rigorous theories and extensive experiments on two real-world datasets demonstrate the theoretical and experimental guarantee on our methods in learning treatment responders classification rules with low regret at the cost of limited abstention.

## 1 Introduction

In a variety of fields where personalized decision-making are of interest, a key problem is to learn individual treatment rules to determine which individuals should be assigned treatments based on individual-level causal effects. To incorporate the population heterogeneity for more individualized and customized decision-making procedure, the target is to learn a function from a rich collection of individual covariates that predicts which treatment should be assigned to the individual. Comparing to classical decision-making tasks, a key challenge in learning individual treatment rule is that we can only observe the factual outcomes corresponding to the treatments are actually assigned. On the other hand, the counterfactual outcome can never be observed, making the treatment effects especially those on individual level hard to identify in practice.

Treatment responder classification aims at learning a classification rule on treatment responders, i.e. individuals with a positive effect from the intervention of interest (Kallus, 2019; 2023). The necessity of classifying treatment responders has been widely addressed in applications (Atkinson et al., 2019; Gloster et al., 2020). For example, in personalized medicine, doctors want to prescribe drugs to individuals who will recover only with the drug taken. In recommendation system, practitioners want to push advertisement to the customers that would purchase the product had they been exposed to the advertisement, and would not purchase had the advertisement not been pushed. While existing frameworks on treatment responder classification even learning individual treatment rules require a deterministic decision be made for each individual, this can lead to significant error on some samples with high uncertainty. For this sake, decision-makers want the rule to opt out of making a decision when the confidence in classification is insufficient, to benefit more precise decision-making on remaining individuals with deterministic classification provided. We refer to such action as **abstention**, i.e. the rule abstains or rejects making decision for some individuals for further investigation. We provide the following example to illustrate the motivation of abstention, with an additional example provided in Appendix B.

**Clinical decision-making in existence of diagnostic ambiguity.** In clinical applications such as chronic and brain-related diseases, doctors want to prescribe effective but risky treatment such as invasive surgery to patients whose disease status is bound to benefit from such surgery, i.e., be a treatment responder. While it is not hard to classify treatment responder for some patients with good physical condition or obvious risk factors, doctors may observe rare lesion for some patients, and the uncertainty on these patients to benefit from the surgery may be significantly higher than

others since there has been few samples with such lesion. Abstention learning, in this case, enables practitioners to select these patients for further actions such as resorting to external information. By withholding judgment on these ambiguous cases for further information, abstention improves the overall accuracy of decisions for patients while saving the cost of expensive clinical testing, enhancing both patient safety and the efficiency of surgical planning.

While abstention learning has been studied in prediction and classification tasks (Cortes et al., 2016; Shekhar et al., 2019), it is not straightforward to extend to the causal scenarios as it faces the following key challenges. First, since treatment responder is determined by both factual and counterfactual outcomes, the identification of the loss and uncertainty terms requires extra assumption and derivation. Second, the identification of loss function in causal setting involves a series of nuisance parameters, which may affect the accuracy of learned treatment rule. This motivates us to develop robust learning methods to deal with possible mis-specification on nuisance parameters. However, prior robust methods in individual treatment rule learning or causal inference did not consider abstention as an option, while existing abstention learning methods also lack theoretical analysis in a causal context. Moreover, traditional classification with abstention typically abstain based on a predefined cost of rejection, which may be hard to interpret in practice (Cortes et al., 2016; Mao et al., 2024). Incorporating treatment responder classification with other types of abstention helps increase the interpretability and broaden the application scenarios of the framework. More discussions on related works are provided in Appendix B.

Being aware of the challenges above, we develop a framework to classify treatment responders with abstention. **The framework mainly consists of a predictor for classification and a rejector to abstain samples.** We start by formulating the misclassification risk using potential outcomes, and provide a constraint-based definition on the targeted abstention rule for enhanced interpretability in application. By revealing the relation between causal misclassification risk with abstention and Conditional Value at Risk (CVaR), a value widely studied in economics. By leveraging the properties of CVaR, we develop a doubly robust method **T**reatment **RE**sponder **C**lassification with **A**bstention (**TRECA**) to learn the classification rule through identification and estimation on the loss function under monotonicity assumption. Taking account of possible violation in monotonicity assumption, we propose a modified method **TRECA**$_+$ to learn the predictor through minimizing tight upper bound derived on the loss function under partial identification, and we also discuss the extension under possible violation on unconfoundedness using sensitivity analysis. Supportive theories have been developed to guarantee the convergence on loss function under loose convergence of nuisance parameters and the accuracy of learned rejector, with extensive experiments on two real-world datasets demonstrate the performance on our methods in classifying treatment responders in a variety of complex scenarios. Our main contributions are summarized as follows:

- To our best knowledge, this is the first work considering causal decision making when abstention is optional, greatly extending the application of treatment responder classification. It also reveals the relation between abstention and CVaR, which can serve as a springboard to inspire further studies on other causal decision making methods with abstention.

- The paper proposes a comprehensive framework to classify treatment responder in various scenarios with and without monotonicity assumption, derives improved partial identification bounds, and proposes robust estimators on the loss function with proper theoretical guarantees.

- Extensive experiments on real-world datasets demonstrate the superiority of our method in classifying treatment responders under varying abstention rates.

## 2 PROBLEM SETUP

We start by introducing the basic setup as well as key notations throughout the paper, with a summary table provided in Appendix C. Consider a group of $n$ units sampled from superpopulation $\mathcal{P}$. Each unit with covariates $X \in \mathcal{X}$ where $\mathcal{X}$ is a bounded vector space receives a binary treatment $T \in \{-1, 1\}$ and produce a binary outcome $Y \in \{-1, 1\}$. Let $Y(1)$ and $Y(-1)$ be the potential outcome under treatment $T = 1$ and $T = -1$, and $Y$ be the observed outcome. In practice, $Y$ can be a binary categorization from a continuous outcome indicating whether the outcome is favored. The individuals are divided into two classes: treatment responders with $Y(1) > Y(-1)$ and non-responders with $Y(1) \leq Y(-1)$. For notation simplicity, let $R = 2I(Y(1) > Y(-1)) - 1$ indicate whether an individual is a treatment responder with $R = 1$

or non-responder with $R = -1$. Let $\rho(X) = \mathbb{P}(R = 1 \mid X)$ be the conditional responding probability, and $\tau(x) = \mathbb{E}[Y(1) - Y(-1) \mid X = x]$ be the conditional average treatment effect (CATE). Denote the propensity score $\pi(X) = \mathbb{P}(T = 1|X)$ and conditional outcome expectation $\mu_t(x) = \mathbb{E}[Y \mid T = t, X = x]$, $\mu(x) = (\mu_1(x), \mu_{-1}(x))$.

Our framework consists of learning a predictor $f : \mathcal{X} \to \{-1, 1\}$ that predicts whether an individual is a treatment responder when $f(x) = 1$ or non-responder when $f(x) = -1$ based on the covariates, as well as an abstention rule $r : \mathcal{X} \to \{0, 1\}$ which abstains making decision on the sample with $X = x$ when $r(x) = 1$ while retain the sample with $r(x) = 0$. Let $\mathcal{F}$ and $\mathcal{R}$ be the hypothesis set of $f$ and $r$ respectively. For $\delta \in (0, 1)$, our goal is to the $\delta$-optimal rule defined as follows

$$(f^*, r^*) = \arg \min_{(f,r) \in \mathcal{F} \times \mathcal{R}} \tilde{L}_\theta(f, r) \text{ s.t. } \mathbb{E}\{r(X)\} \leq \delta, \tag{1}$$

where

$$\tilde{L}_\theta(f, r) = \theta \cdot \mathbb{P}(f(X) = 1, R = -1, r(X) = 0) + (1 - \theta) \cdot \mathbb{P}(f(X) = -1, R = 1, r(X) = 0).$$

Term $\tilde{L}_\theta(f, r)$ is the misclassification loss predicting $R$ from $X$ on samples that have not been abstained. Hyper-parameter $\theta$ weights the importance between false positive and false negative samples to match the requirement in practice. For example, in the classification on drug responders, doctors may prefer false negative to false positive samples for safety concern. The condition $\mathbb{E}\{r(X)\} \leq \delta$ ensures that the abstention rule abstain less than $\delta$-proportion of samples on the superpopulation $\mathcal{P}$ to ensure its generalization performance. Comparing to the cost-based abstention where the choice of abstention is assumed to induce a prespecified cost (Cortes et al., 2016; Mao et al., 2024), the constraint-based abstention rule that determines the proportion of abstained units avoids explicit definition on the reject cost, thus is more interpretable in many application scenarios (Franc et al., 2023). In the special case when $\delta = 0$, i.e., no samples being abstained, $\tilde{L}_\theta(f, r)$ degenerates to the misclassification loss discussed in previous works on treatment responder classification such as Kallus (2019); Wu et al. (2025). Our basic framework is built upon the following classical assumptions in causal inference:

**Assumption 1 (Consistency)** *The observed outcome equals to the potential outcome under assigned treatment, i.e. $[Y_i \mid T_i = t] = Y_i(t)$ for any unit and $t \in \mathcal{T}$.*

**Assumption 2 (Positivity)** *There exists $0 < \underline{\pi} < 1$ such that $\forall X \in \mathcal{X}, \pi(x) \in [\underline{\pi}, 1 - \underline{\pi}]$.*

**Assumption 3 (Unconfoundedness[1])** $(Y(0), Y(1)) \perp\!\!\!\perp T|X$.

Intuitively, a good abstention rule that achieves low misclassification loss with limited abstention is an **uncertainty-based** criterion, which rejects samples that are highly uncertain to ensure better classification accuracy on retained samples. In the following discussion, we will provide a theoretically-grounded measurement on the uncertainty in treatment responder classification task using conditional risk and derive its identification under rigorous conditions.

## 2.1 Misclassification Loss under Abstention is Conditional Value at Risk

To start with, we introduce the following conditional risk for uncertainty measurement

$$V_f(x) = \mathbb{E}\{\theta I(f(X) = 1, R = -1) + (1 - \theta)I(f(X) = -1, R = 1) \mid X = x\}.[2] \tag{2}$$

The conditional risk measures the uncertainty on $f$ predicting the true responder status $R$ given covariates. Due to the randomness at individual level, $R$ is not fixed given covariates. Such randomness is also the key difference between individual treatment effect (ITE) and conditional average treatment effect (CATE) (Vegetabile, 2021). Therefore, the conditional expectation taken in (2) originates from the individual level, and high conditional risk indicates high uncertainty on the responder type among individuals given the same covariate observation. In remark, conditional risk is different from the Conditional Value at Risk discussed later. We impose the following minor regularity assumptions on $V_f(X)$.

---

[2]For the notation on conditional risk, we omit $\theta$ in default for simplicity.

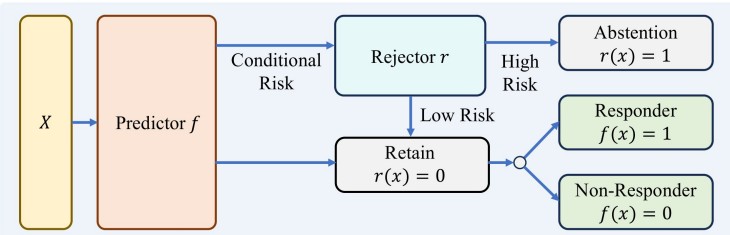

Figure 1: Framework on treatment responder classification with abstention.

**Assumption 4 (Continuity)** $V_f(X)$ *is a random variable with continuously differentiable cumulative probability function* $F_{V_f}$ *with bounded density* $\|F'_{V_f}\|_\infty \le c$.

Assumption 4 assumes that the conditional risk $V_f(X)$ has bounded probability density function. It ensures that the quantile $F_{V_f}^{-1}(1-\delta)$ well defined to ensure uniqueness on the optimal abstention rule. Assumption 4 is also applied to derive robustness theorems.

**Theorem 1** *Under Assumption 1-4, the $\delta$-optimal rule has the following equivalent form:*

$$f^* = \arg\min_{f\in\mathcal{F}} L_{\theta,\delta}(f)$$

*with*

$$L_{\theta,\delta}(f) = \mathbb{E}\left\{\theta I(f(X)=1, R=-1) + (1-\theta)I(f(X)=-1, R=1)|V_f(X) \le F_{V_f}^{-1}(1-\delta)\right\},$$

(3)

*and* $r^*(x) = I(V_{f^*}(X) > F_{V_{f^*}}^{-1}(1-\delta))$. *Moreover, under Assumption 1 and 3, we have*

$$L_{\theta,\delta}(f) = \mathbb{E}\left\{V_f(x) \mid V_f(x) \le F_{V_f}^{-1}(1-\delta)\right\}.$$

(4)

Theorem 1 indicates that the optimal predictor that minimizes the misclassification loss $\tilde{L}_\theta(f,r)$ on retained samples under constraint on abstention rate is equivalent to the minimizer of $L_{\theta,\delta}(f)$ defined in (3). Given the optimal predictor $f^*$, the optimal abstention rule abstains samples by thresholding the $(100\delta)\%$ highest conditional risk. This inspires us to train the predictor and rejector in a sequential order. In the first step, we train the predictor through minimizing the empirical form of $L_{\theta,\delta}(f)$ and estimate the conditional risk for each individual. The abstention rule is thereon determined as a thresholding rule rejecting samples with the highest conditional risk for better accuracy on the retained samples with relatively low conditional risk. A diagram on the overall framework is presented in Figure 1.

The loss function in (4) represents the expectation on the conditional risk $V_f(x)$ among the $(100\delta)\%$ lowest values. Such average among subpopulation with the lowest scores has been explored as the Conditional Value at Risk (CVaR). CVaR commonly appears to be the target of interest in financial engineering or management studies, where the extreme behaviors are of interest (Krokhmal et al., 2002; Sarykalin et al., 2008; Filippi et al., 2020). While multiple works discussed about the optimization, estimation and asymptotic properties on CVaR (Uryasev, 2000; Rockafellar et al., 2000; Chun et al., 2012), in particular, Kallus (2023) developed the statistical inference procedure on CVaR in terms of ITE, studying CVaR in the context of individual treatment rule learning remains little explored. Inspired by Rockafellar et al. (2000), we have the following equivalent form on loss function, which will be leveraged to construct estimators in the next section.

**Proposition 1** *Under Assumption 1-5, the loss function $L_{\theta,\delta}(f)$ defined in (3) has the following equivalent form:*

$$L_{\theta,\delta}(f) = \sup_{\beta\in\mathbb{R}}\left\{\beta + \frac{1}{1-\delta}E(V_f(X)-\beta)_-\right\},$$

*where* $(u)_- = u \wedge 0$, $r^*(x) = I(V_f(X) > \beta^*)$, *and* $\beta^* = \arg\max_{\beta\in\mathcal{R}}\left\{\beta + \frac{1}{1-\delta}\mathbb{E}(V_f(X)-\beta)_-\right\}$.

Proposition 1 formulates the loss function as the supremum over an additional parameter $\beta$. The optimal value $\beta^*$ coincides with the threshold for the abstention rule. In practice, we can either rank the estimates $\{\hat{V}_f(X_i)\}_{i=1}^n$ and abstain the highest $(100\delta)\%$ values to ensure a strict abstention rate $\delta$ or use the optimizer $\hat{\beta}$ as the threshold to enhance efficiency and generalizability. The details for loss estimation and optimization will be discussed in the next section.

## 3 METHODOLOGY AND THEORIES

### 3.1 IDENTIFICATION AND ESTIMATION OF LOSS FUNCTION UNDER MONOTONICITY

From Proposition 1, minimizing the loss function in (3) is estimable once the conditional risk $V_f(x)$ is identified. In light of this, we develop the identification on loss function $L_{\theta,\delta}(f)$ through leveraging the following monotonicity assumption also made in Kallus (2019):

**Assumption 5 (Monotonicity)** $Y(1) \geq Y(-1)$.

As shown in Table 1, monotonicity assumption essentially implies that the subgroup with $Y(1) = -1, Y(-1) = 1$ do not exist. In drug-treatment example, it means that the drug will not have a harmful effect on the patient. While it is commonly made in causal effect identification such as distributional treatment effect (Kim, 2014) and local average treatment effect (Angrist & Imbens, 1995), there are cases that monotonicity does not hold. In Section 3.2 discuss the extension of our method when monotonicity assumption may not hold. The conditional risk is identified by the following proposition:

**Proposition 2** *Under Assumption 1-3*, $V_f(x) = \frac{1}{2}f(x)\{\theta - \rho(x)\} + \frac{1}{2}\{\theta + (1 - 2\theta)\rho(x)\}$, *and* $\rho(x) = \tau(x)/2 + P(Y(1) < Y(-1) \mid X = x)$. *When Assumption 5 holds*, $\rho(x) = \tau(x)/2$.

**Table 1:** Categorization of $(Y(1), Y(-1))$ when monotonicity holds or violates.

| Monotonicity assumption | Hold ($\checkmark$) | May not hold (?) |
|---|---|---|
| Treatment responder | $(+1, -1)$ | $(+1, -1), (-1, +1)$ |
| Treatment non-responder | $(+1, +1), (-1, -1)$ | $(+1, +1), (-1, -1)$ |

We write $V_f(x; \tau)$ substituting $\rho(x)$ in $V_f(x)$ with $\tau(x)/2$ to address that the identification involves $\tau$. The identification in Proposition 2 introduces CATE $\tau(x)$ as nuisance parameter that affects the performance of targeted estimator. The concern on nuisance parameters has also been aroused in a wide range of causal studies, including effect estimation, policy learning and causal machine learning (Bang & Robins, 2005; Chernozhukov et al., 2018; Moosavi et al., 2023). A straightforward way to optimize $L_{\theta,\delta}(f)$ in practice is to minimize its naive empirical estimator $\hat{L}_{\theta,\delta}^{naive}(f; \hat{\tau}) = \sup_{\beta \in \mathbb{R}} \left\{ \beta + \frac{1}{n(1-\delta)} \sum_{i=1}^n (V_f(X_i; \hat{\tau}) - \beta)_- \right\}$, where $V_f(x; \hat{\tau})$ is the plug-in estimator of $V_f(x; \tau)$. However, as the loss is sensitive to the estimation on $\tau$ as well as $\beta$, when $\hat{\tau}$ or $\hat{\beta}$ is non-negligibly biased or converges slowly, $\hat{L}_{\theta,\delta}^{naive}(f; f, \hat{\tau})$ may not convergence to $L_{\theta,\delta}(f)$ at a decent rate. Taking account of this, we construct a robust estimation on loss function that is less sensitive to the accuracy of $\hat{\tau}(x)$ as follows. Then the estimator for loss function is constructed as

$$\hat{L}_{\theta,\delta}(f; \hat{\tau}, \hat{\mu}, \hat{\pi}) = \sup_{\beta \in \mathbb{R}} \left\{ \frac{1}{n} \sum_{i=1}^n l_{\theta,\delta}(X_i, T_i, Y_i, f; \hat{\tau}, \hat{\mu}, \hat{\pi}, \beta) \right\}, \tag{5}$$

where

$$l_{\theta,\delta}(X, T, Y, f; \tau, \mu, \pi, \beta) = \beta + \frac{1}{1-\delta}I(V_f(X; \tau) \leq \beta)\Big[\frac{1}{2}(f(X) + 1)\theta + \frac{1}{4}\{1 - 2\theta - f(X)\} \times$$
$$\left\{ \mu_1(X) - \mu_{-1}(X) + \frac{(T+1)/2 - \pi(X)}{\pi(X)(1 - \pi(X))}(Y - \mu_T(X)) \right\} - \beta \Big].$$

The augmented loss function $l_{\theta,\delta}$ is constructed by substituting CATE $\tau(X)$ in $V_f(X; \tau) - \beta$ with a doubly robust estimator modeled by $\mu$ and $\pi$ (Bang & Robins, 2005; Chernozhukov et al., 2018). The empirical loss function $\hat{L}_{\theta,\delta}(f; \hat{\tau}, \hat{\mu}, \hat{\pi})$ is the plug-in estimator of $l_{\theta,\delta}$ with nuisance parameters

$(\tau, \mu, \pi, \beta)$. Recall that $\beta^* = \arg\max_{\beta \in \mathbb{R}} l_{\theta,\delta} \mathbb{E}(X, T, Y, f; \tau, \mu, \pi, \beta)$, with $(\tau, \mu, \pi)$ be the true functions. Comparing with the naive estimator $\hat{L}_{\theta,\delta}^{naive}(f; \hat{\tau})$, $\mathbb{E}l_{\theta,\delta}(X, T, Y, f; \hat{\tau}, \hat{\mu}, \hat{\pi}, \hat{\beta})$ has zero partial derivative at $(\tau, \mu, \pi, \beta^*)$, which also known as Neyman orthogonality condition. This condition ensures that the expectation on $l_{\theta,\delta}$ as well as its empirical mean is not sensitive to small error on nuisance parameters near the true value, thus ensuing double robustness property (Kallus, 2023). To state formally, the following theorem illustrates the double robustness property.

**Theorem 2** *Suppose Assumption 1-5 hold, and $V_f(x; \hat{\mu})$ has density function bounded by c. There exists $\tilde{\pi} \in [\underline{\pi}, 1 - \underline{\pi}]$ such that the estimators for nuisance parameters satisfy $\|\hat{\pi} - \tilde{\pi}\| = o_p(1)$, $\|\hat{\mu} - \tilde{\mu}\| = o_p(1)$, and $\|\hat{\tau} - \tau\|_q = O_p(\epsilon_n^{\frac{q}{q+1}})$ for some $q > 1$. Then when either of the following conditions holds:*
$$\|\hat{\pi} - \pi\| = o_p(\epsilon_n) \qquad \|\hat{\mu} - \mu\| = o_p(\epsilon_n),$$
*we have $|\hat{L}_{\theta,\delta}(f; \hat{\tau}, \hat{\mu}, \hat{\pi}) - L_{\theta,\delta}(f)| = O_p(\epsilon_n \vee n^{-1/2})$.*

Theorem 2 suggests that even if $\pi$ and $\mu$ are inconsistent, we can get a consistent estimator on loss function once we have a consistent estimator, although it can converge slowly, on $\tau$. Note that while $\tau(x) = \mu_1(x) - \mu_{-1}(x)$, $\mu$ and $\tau$ are treated as separated parameters which means that they do not require the same model. For instance, $\hat{\tau}$ can be estimated through direct CATE models without modeling the conditional expectations such as causal forests (Wager & Athey, 2018) or R-learner (Nie & Wager, 2021). The proof on Theorem 2 is provided in the Appendix. Note that from the proof details, the error bound is linear to the constants $c$, $(1 - \delta)^{-1}$ and $\underline{\pi}^{-1}$. Therefore, the convergence rate is not sensitive to the problem-specific constants. We also derive generalization bound on $\hat{L}_{\theta,\delta}(f; \hat{\tau}, \hat{\mu}, \hat{\pi})$ by measuring space complexity through Rademacher complexity, see Theorem 5 in appendix for details.

### 3.2 DISCUSSION WHEN MONOTONICITY DOES NOT HOLD

As mentioned before, although monotonicity assumption is commonly made in identification of causal parameters, there are real-world scenarios where it may violate. In this section, we consider the case in which $V_f(x)$ as well as loss function $L_{\theta,\delta}(f)$ can not be fully identified when monotonicity assumption may not hold. Instead, we seek to minimize an upper bound on loss function. Recall the expression and definition on $\beta^*$ in Proposition 1, if we can find a tight upper bound $U_f(x)$ on conditional risk, i.e. $U_f(x) \geq V_f(x)$ almost everywhere, then $M_{\theta,\delta}(f) := \sup_{\beta \in \mathbb{R}} \left\{ \beta + \frac{1}{1-\delta} \mathbb{E}(U_f(x) - \beta)_- \right\}$ is a tight upper bound on $L_{\theta,\delta}(f)$ since

$$M_{\theta,\delta}(f) \geq \beta^* + \frac{1}{1-\delta} E(U_f(x) - \beta^*)_- \geq \beta^* + \frac{1}{1-\delta} E(V_f(X) - \beta^*)_- = L_{\theta,\delta}(f).$$

**Proposition 3** *For functions $\rho^L(x), \rho^U(x)$ such that $\rho(x) \in (\rho^L(x), \rho^U(x))$ almost everywhere, we have the following tight upper bound on conditional risk*

$$V_f(x) \leq \frac{1 - f(x)}{2} \cdot (1 - \theta)\rho^U(x) + \frac{1 + f(x)}{2} \cdot \theta(1 - \rho^L(x)). \tag{6}$$

Proposition 3 shows that to construct upper bound on conditional risk $V_f(x)$, it suffices to derive bounds to partially identify $\rho(x)$. The proposed bound (6) improves Theorem 4.4 in (Wu et al., 2025) in the sense that given $f(x) \in \{-1, 1\}$, equality in (6) is attained once $\rho(x) = \rho^U(x)$ or $\rho(x) = \rho^L(x)$ but not necessarily both. From Proposition 2, $\rho(x) = \frac{\tau(x)}{2} + \mathbb{P}(Y(1) < Y(-1) \mid X = x)$. The later term $\mathbb{P}(Y(1) < Y(-1) \mid X = x)$, has been explored in previous works as the fraction negatively affected (FNA) (Kallus, 2022). In this paper, we adopt the upper and lower bounds in the form $\rho^U(x) = u(\tau(x))$ and $\rho^L(x) = w(\tau(x))$, where $u$ and $w$ are two known functions that are differentiable with regard to $\tau$. For example, from Li et al. (2023); Wu et al. (2025), we can take $w(\tau) = \max\{\frac{\tau}{2}, 0\}$, and $u(\tau) = \frac{1}{2} + \frac{\tau}{4}$. Under assumption that $Y(1), Y(-1)$ are non-negatively correlated given $X$, the upper bound can be tightened to $u(\tau) = (\frac{1}{2} + \frac{\tau}{4})^2$.

In conclusion, let $U_f(x; \tau)$ be the upper bound on conditional risk derived in (3). The debiased estimator for upper bound on loss function $M_{\theta,\delta}(f)$ is constructed as

$$\hat{M}_{\theta,\delta}(f; \hat{\tau}, \hat{\mu}, \hat{\pi}) = \sup_{\beta \in \mathbb{R}} \left\{ \frac{1}{n} \sum_{i=1}^n m_{\theta,\delta}(X_i, T_i, Y_i, f; \hat{\tau}, \hat{\mu}, \hat{\pi}, \beta) \right\}, \tag{7}$$

$$m_{\theta,\delta}(X,T,Y,f;\tau,\mu,\pi,\beta) = \beta + \frac{1}{1-\delta}I(U_f(X;\tau) \leq \beta)\Big[U_f(X;\tau) - \beta$$
$$+ \partial_\tau U_f(X;\tau)\frac{(T+1)/2 - \pi(X)}{\pi(X)(1-\pi(X))}(Y - \mu_T(X))\Big],$$

where $\partial_\tau U_f(X;\tau) = \frac{(1-f(x))(1-\theta)}{2}u'(\tau(x)) - \frac{(1+f(x))\theta}{2}w'(\tau(x))$, $w', u'$ are the derivatives of $w, u$. Note that $\mathbb{E}(U_f(x) - \beta)_- = \mathbb{E}I(U_f(x) \leq \beta)(U_f(x) - \beta)$, the augmented loss function $m_{\theta,\delta}$ appends a re-weighted doubly robust residual to the latter term $U_f(X;\tau) - \beta$. While such residual has zero mean when either of $\mu, \pi$ is specified, the double robustness property still holds by check the orthogonality of nuisance parameters $(\hat{\tau}, \hat{\beta})$ on $\mathbb{E}m_{\theta,\delta}(X,T,Y,f;\hat{\tau},\mu,\pi,\hat{\beta})$. The double robustness property is rigorously formulated in Theorem 3.

**Theorem 3** *Suppose Assumption 1-4 hold, and we assume $U_f(x;\hat{\tau})$ has bounded density instead. Then for estimators $(\hat{\tau}, \hat{\mu}, \hat{\pi})$ satisfying the same convergence and doubly robustness conditions as in Theorem 2, we have $|\hat{M}_{\theta,\delta}(f;\hat{\tau},\hat{\mu},\hat{\pi}) - M_{\theta,\delta}(f)| = O_p(\epsilon_n \vee n^{-1/2})$.*

Finally, to provide further theoretical guarantee on the performance of abstention rule learned under partial identification, we develop Theorem 4 to bound the rate on misclassification probability of the abstention rule learned from the upper bound on loss function derived under partial identification $\rho(x) \in (\rho^L(x), \rho^U(x))$. Let $\hat{\rho}^U(x) = u(\hat{\tau}(x))$, $\hat{\rho}^L(x) = w(\hat{\tau}(x))$ be the estimators on sensitivity bound, and let $d_\infty = \max_{x \in \mathcal{X}}|\hat{\rho}^U(x) - \hat{\rho}^L(x)|$ denote the largest length over $x$.

**Theorem 4** *Given estimators $(\hat{\tau}, \hat{\mu}, \hat{\pi})$ satisfying Assumption 1-4 and conditions in Theorem 2, let $\hat{f}_M = \arg\min_{f \in \mathcal{F}} \hat{M}_{\theta,\delta}(f;\hat{\tau},\hat{\mu},\hat{\pi})$ be the treatment rule learned from the upper bound on classification loss and $r^M(x)$ be the estimated abstention rule. Assume that (a) $\rho(X)$ has bounded density function, and (b) $\hat{f}_M(x) = -1$ for $\rho^U(x;\hat{\tau}) \leq \theta$ and $\hat{f}_M(x) = 1$ for $\rho^L(x;\hat{\tau}) > \theta$, we have*

$$\mathbb{P}(\hat{r}_M(X) \neq r^*(X)) = O(d_\infty \vee n^{-1/2}).$$

Theorem 4 shows that for sufficiently large hypothesis set, narrower the partial identification bound is, more accurate the learned rule will be. In particular, when $\rho(x)$ is fully identified, i.e., $d_\infty = 0$, we can learn the $\delta$-optimal abstention rule at an error rate of at most $n^{-1/2}$. Condition *(b)* assumes that the best estimated predictor can classify the responder correctly for deterministic samples under partial identification. As a special case, condition *(b)* is satisfied when the VC dimension of hypothesis set $\mathcal{F}$, $\text{VC}(\mathcal{F}) \geq n$ such that the sample points can be shattered by the models in $\mathcal{F}$. The hypothesis set $\mathcal{F}$ with sufficiently large VC dimension can be achieved through various machine learning models such as neural networks. For instance, for ReLU networks, it is shown that $\text{VC}(\mathcal{F}) \lesssim WL\log(W/L)$, where $W$ and $L$ are the number of nodes and layers (Bartlett et al., 2019). Although such a large hypothesis set may induce overfitting, it serves an illustrative example to depict the performance of our method under partial identification on $\rho(x)$ without sculpturing the hypothesis set.

### 3.3 DISCUSSION WHEN UNCONFOUNDEDNESS DOES NOT HOLD

In this section, we extend the method in existence of unmeasured confounders. The key problem is that $\tau(x)$ can not be fully identified. We deal with it by making sensitivity analysis on CATE and derive bounds on conditional risk $V_f(x)$. Suppose the sensitivity bound constructed on $\tau(x)$ is $\tau(x) \in [\tau^L(x), \tau^U(x)]$. Then from Proposition 2, under Assumption 1, 2 and 4, $\rho(x) \in [\tau^L(x)/2, \tau^U(x)/2]$. Therefore, from Proposition 3, we have

$$V_f(x) \leq U_f(X) := \frac{1 - f(x)}{2} \cdot (1 - \theta)\frac{\tau^U(x)}{2} + \frac{1 + f(x)}{2} \cdot \theta(1 - \frac{\tau^L(x)}{2})$$

and hence we have $L_{\theta,\delta}(f) \leq \sup_{\beta \in \mathbb{R}}\left\{\beta + \frac{1}{1-\delta}\mathbb{E}(U_f(X) - \beta)_-\right\}$. The treatment responder is selected through minimizing the upper bound above with the same procedure as in Algorithm 1. In practice, we adopt the plug-in estimator on the following sensitivity bound inspired by the sensitivity

bound discussed in Section 22.3 in Imbens & Rubin (2015):

$$\tau^U(x) = \pi(x)\mu_{1,1}(x) + 1 - [1 - \pi(x)]\mu_{-1,-1}(x);$$
$$\tau^L(x) = \pi(x)\mu_{1,1}(x) - 1 - [1 - \pi(x)]\mu_{-1,-1}(x),$$

where $\mu_{t,t'}(x) = \mathbb{E}(Y(t) \mid T = t', X = x)$. In the experiment results, we will demonstrate that the extended version of TRECA performs well comparing with baseline methods in making more accurate classification on treatment responders among retained samples.

### 3.4 TIME AND SPACE COMPLEXITY ANALYSIS

**Time complexity.** Let $n$ be the sample size, $m$ the batch size, and let $l_\pi, l_\mu, l_\tau, l_f$ and $d_\pi, d_\mu, d_\tau, d_f$ denote the number of layers and hidden dimensions for the propensity, outcome, CATE, and predictor networks, respectively. Let $t_\phi = l_\phi d_\phi^2$ represent the time complexity of a one-time forward pass in network $\phi$. The analysis of the time complexity of our method can be divided into three stages. In Stage 1, $\hat{\pi}(x)$ and $\hat{\mu}_t(x)$ are trained by minimizing the MSE over the entire samples, which results in a complexity of $O(n(t_\pi + 2t_\mu))$. In Stage 2, we train the CATE $\tau$ and predictor $f$ alternatively by minimizing the CVaR loss. Estimating the CVaR loss requires calling $\hat{\pi}(x)$ and $\hat{\mu}_t(x)$, which is performed $r_\beta$ rounds for each of the $n/m$ epochs, resulting in a time complexity of $O(nm^{-1}r_\beta(t_\pi + t_\mu))$. The CATE and predictor are then trained with complexity $O(n(t_\tau + t_f))$. Balancing the representations in CATE estimation involves a time complexity of $O(n/m \cdot m^2) = O(nm)$. In Stage 3, the conditional risks $V_{\hat{f}}(x)$ are computed for each individual using the trained predictor, with time complexity $O(n(t_\tau + t_f))$. In summary, the overall time complexity is $O(n[2(t_\tau + t_f) + (1 + m^{-1}r_\beta)(d_\pi + d_\mu) + m])$, which is linear in all quantities, implying that the method generally does not suffer from heavy computational burden.

**Space complexity.** Following the notations and stage division in time complexity, further define $s_\phi = d_\phi l_\phi$ to be the capacity of some network $\phi$. In Stage 1, the space complexity is equal to the network capacity times batch size $O(m(s_\pi + s_\mu))$. In Stage 2, since the model is trained through alternative gradient descent on $\tau$ and $f$ while requires calling the propensity for each individual, the space complexity for weight storage, gradient memory and propensity storage is $O(m(s_\tau + s_f))$. Computing the representation distance in learning CATE involves $O(m^2)$ space. Learning the abstention rule in Stage 3 requires $O(1)$ extra space. In summary, the space complexity is $O(m(s_\pi + s_\mu + s_\tau + s_f) + m^2)$, which is at most quadratic to relevant parameters. In Appendix E.4, experiments indicate that **TRECA** has a medium level of execution time and memory comparing with baseline methods, providing a balance between efficiency and accuracy.

## 4 EXPERIMENTAL RESULTS

### 4.1 EXPERIMENTAL SETUP

**Datasets and Preprocessing.** To measure the effectiveness of the proposed methods, we conduct extensive experiments on two real-world datasets, **Twins** (Almond et al., 2005) and **Jobs** (LaLonde, 1986) and their transformed datasets **Twins_mono** and **Jobs_mono** to ensure monotonicity. See Appendix E for detailed description on datasets, preprocessing procedure and supplementary results. [3]

**Baselines.** In the context of causal classification, we compare our results with the treatment rules learned from the following methods. (1) Meta-learners, which combines multiple models to enhance performance, such as **X-learner** and **T-learner** (Künzel et al., 2019; Salditt et al., 2024). (2) Counterfactual Representation Learning (CFR) based models, including **CFRNet** (Shalit et al., 2017), **DeRCFR** (Wu et al., 2022), **ESCFR**(Wang et al., 2023), **DragonNet** (Shi et al., 2019) and **CFRISW** (Hassanpour & Greiner, 2019). (3) Other methods, including tree-based method Causal Forest **CF** (Athey & Imbens, 2016), **CEVAE** (Louizos et al., 2017), which uses variational inference to generate counterfactual outcomes. We also compare our methods with **CTRL** (Wu et al., 2025) which deals with treatment responder classification without taking account of abstention.

---

[3]Also see anonymous link `https://anonymous.4open.science/r/TRECA-4A0F` for code.

**Evaluation Metrics.** We use the False Negative Rate (**FNR**), False Positive Rate (**FPR**), and the regret on predictor $f$ (**f-regret**) to evaluate the performance of each methods. In the experiment, we randomly abstain the samples on baselines to ensure the metrics are computed on the same amount of samples. We also compare our method with baseline methods under uncertainty-based abstention strategy, with details provided in Appendix E.4. Denote $R_i$ as the responder indicator for individual $i$. For sample set $\mathcal{M}$ with size $m$, the metrics on $\mathcal{M}$ are computed by $\textbf{FNR} = \sum_{i \in \mathcal{M}} I(\hat{f}(X_i) = -1, R_i = 1, \hat{r}(X_i) = 0)/|\mathcal{M}|$, $\textbf{FPR} = \sum_{i \in \mathcal{M}} I(\hat{f}(X_i) = 1, R_i = -1, \hat{r}(X_i) = 0)/|\mathcal{M}|$ and $\textbf{f-regret} = \theta \cdot \textbf{FNR} + (1 - \theta) \cdot \textbf{FPR}$ is a weighted combination. We take $\theta = 0.5$ throughout the experiment, and evaluate the within-sample and out-of-sample performances by replicating training and testing sets respectively with five rounds to report the mean and standard deviation.

**Model Setup.** In both **TRECA** and **TRECA$_+$**, we use CFRNet (Shalit et al., 2017) in training the CATE estimator $\hat{\tau}$ and condition expectation $\hat{\mu}$. MLP network is applied to model the predictor $\hat{f}$ and propensity score $\hat{\pi}$. To derive gradient on loss function, we use the Softplus function $log(1+exp(x))$ as surrogate to the indicator function. More details on the configuration and training process are discussed in Appendix E.1.

### 4.2 Experimental Results

Table 2 compares the baseline models with **TRECA** and **TRECA$_+$**, the versions of our method considering and not considering monotonicity assumption, with details described in Algorithm 1 and 2. The results indicate that (1) our method **TRECA** stably outperforms baseline methods in **Jobs_mono** and most cases in **Twins_mono**, which can result from two main advantages of our method. First, our method directly minimizes the misclassification loss, enabling us to learn a more accurate classification rule comparing with estimation-then-decision rules derived from CATE-oriented methods (Fernández-Loría & Provost, 2022). Second, **TRECA** is derived to under the monotonicity assumption (Assumption 5). With real data matching the assumption, such extra information leads to a more accurate classification. (2) Both methods **TRECA** and **TRECA$_+$** have comparatively small standard deviation. Such observation can be attributed to the efficiency in the doubly robust property of our method,

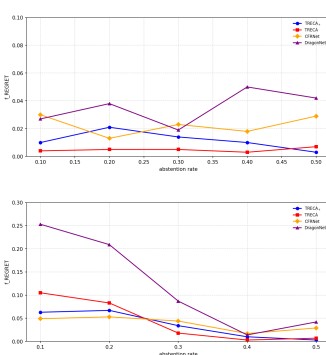

Figure 2: Results of f-regret on Twins_mono and Jobs_mono datasets with varying abstention rate $\delta$.

as it has been shown that doubly robust estimators reach semi-parametric efficiency bound for various causal effect models (Kennedy, 2024). (3) While **TRECA$_+$** does not require monotonicity, the method still outperforms most methods in retained samples, implying its robust performance in various scenarios. (4) The CFR-based methods, e.g. **CFRNet**, **CFRISW**, show high competitiveness among baseline methods, which validates our choice of CFRNet on the model of $\hat{\tau}$ and $\hat{\mu}$ in practice.

Table 3: Results comparison (mean±std) of **TRECA$_+$**, **TRECA CFRNet** on real-world datasets with abstention rate = 30%

| Method | TRECA$_+$ | TRECA | CFRNet |
|---|---|---|---|
| Twins | 0.038 ± 0.045 | **0.006 ± 0.005** | 0.023 ± 0.072 |
| Twins_mono | 0.027±0.020 | **0.004 ± 0.002** | 0.060 ± 0.145 |
| Jobs | 0.111 ± 0.156 | **0.036 ± 0.039** | 0.373 ± 0.195 |
| Jobs_mono | 0.061 ± 0.052 | **0.019 ± 0.016** | 0.179 ± 0.286 |

### 4.3 Ablation Studies

We conduct comprehensive studies to evaluate each component of our method. In the first study, we evaluate the performances of our methods in the original **Twins** and **Jobs** dataset, when monotonicity violates. Table 3 shows that both **TRECA** and **TRECA$_+$** still induce comparatively small f-regret when monotonicity violates. In Jobs dataset, **TRECA$_+$** outperforms **TRECA** a bit with the reverse result on Twins, implying the adaptivity of both of our methods to non-monotonous datasets. The

**Table 2:** Results (mean±std) of FPR, FNR, f-regret on Twins_mono/Jobs_mono datasets with abstention rate = 30%

| Method | Within-samples on Twins_mono | | | Out-of-samples on Twins_mono | | |
| --- | --- | --- | --- | --- | --- | --- |
| | FPR ↓ | FNR ↓ | f-regret ↓ | FPR ↓ | FNR ↓ | f-regret ↓ |
| T-learner | $0.032 \pm 0.165$ | $0.009 \pm 0.002$ | $0.020 \pm 0.082$ | $0.030 \pm 0.159$ | $0.007 \pm 0.003$ | $0.019 \pm 0.079$ |
| X-learner | $0.051 \pm 0.185$ | $0.009 \pm 0.002$ | $0.030 \pm 0.092$ | $0.050 \pm 0.183$ | $0.008 \pm 0.004$ | $0.029 \pm 0.091$ |
| CFRNet | $0.035 \pm 0.149$ | $\mathbf{0.001 \pm 0.002}$ | $0.018 \pm 0.075$ | $0.039 \pm 0.145$ | $0.007 \pm 0.004$ | $0.023 \pm 0.072$ |
| CEVAE | $0.082 \pm 0.065$ | $0.007 \pm 0.002$ | $0.045 \pm 0.032$ | $0.089 \pm 0.061$ | $0.006 \pm 0.004$ | $0.047 \pm 0.030$ |
| ESCFR | $0.041 \pm 0.180$ | $0.001 \pm 0.002$ | $0.021 \pm 0.090$ | $0.048 \pm 0.182$ | $0.007 \pm 0.004$ | $0.027 \pm 0.090$ |
| DeRCFR | $0.029 \pm 0.162$ | $0.001 \pm 0.003$ | $0.015 \pm 0.081$ | $0.039 \pm 0.165$ | $0.007 \pm 0.004$ | $0.023 \pm 0.082$ |
| DESCN | $0.060 \pm 0.240$ | $0.002 \pm 0.004$ | $0.031 \pm 0.120$ | $0.066 \pm 0.238$ | $0.006 \pm 0.004$ | $0.036 \pm 0.118$ |
| DragonNet | $0.132 \pm 0.313$ | $0.002 \pm 0.003$ | $0.067 \pm 0.156$ | $0.137 \pm 0.311$ | $0.006 \pm 0.004$ | $0.071 \pm 0.155$ |
| CF | $0.011 \pm 0.041$ | $0.001 \pm 0.002$ | $0.006 \pm 0.021$ | $0.019 \pm 0.042$ | $0.008 \pm 0.003$ | $0.013 \pm 0.021$ |
| CFRISW | $\mathbf{0.003 \pm 0.001}$ | $0.009 \pm 0.002$ | $0.006 \pm 0.001$ | $\mathbf{0.003 \pm 0.002}$ | $0.008 \pm 0.003$ | $0.005 \pm 0.003$ |
| CTRL | $0.024 \pm 0.052$ | $0.004 \pm 0.003$ | $0.014 \pm 0.026$ | $0.027 \pm 0.059$ | $0.007 \pm 0.003$ | $0.017 \pm 0.029$ |
| **TRECA$_+$** | $0.020 \pm 0.011$ | $0.004 \pm 0.003$ | $0.012 \pm 0.005$ | $0.023 \pm 0.014$ | $\mathbf{0.004 \pm 0.003}$ | $0.014 \pm 0.007$ |
| **TRECA** | $0.003 \pm 0.005$ | $0.007 \pm 0.001$ | $\mathbf{0.005 \pm 0.002}$ | $0.003 \pm 0.005$ | $0.006 \pm 0.003$ | $\mathbf{0.005 \pm 0.003}$ |

| Method | Within-samples on Jobs_mono | | | Out-of-samples on Jobs_mono | | |
| --- | --- | --- | --- | --- | --- | --- |
| | FPR ↓ | FNR ↓ | f-regret ↓ | FPR ↓ | FNR ↓ | f-regret ↓ |
| T-learner | $0.227 \pm 0.153$ | $0.048 \pm 0.059$ | $0.119 \pm 0.051$ | $0.220 \pm 0.153$ | $0.040 \pm 0.051$ | $0.112 \pm 0.050$ |
| X-learner | $0.195 \pm 0.164$ | $0.043 \pm 0.046$ | $0.104 \pm 0.058$ | $0.186 \pm 0.164$ | $0.037 \pm 0.042$ | $0.097 \pm 0.059$ |
| CFRNet | $0.033 \pm 0.136$ | $0.040 \pm 0.016$ | $0.037 \pm 0.051$ | $0.042 \pm 0.139$ | $0.046 \pm 0.022$ | $0.044 \pm 0.053$ |
| CEVAE | $0.064 \pm 0.075$ | $0.030 \pm 0.040$ | $0.047 \pm 0.045$ | $0.090 \pm 0.075$ | $0.025 \pm 0.026$ | $0.058 \pm 0.051$ |
| ESCFR | $0.012 \pm 0.015$ | $0.055 \pm 0.030$ | $0.040 \pm 0.034$ | $0.015 \pm 0.017$ | $0.054 \pm 0.030$ | $0.035 \pm 0.025$ |
| DeRCFR | $0.050 \pm 0.154$ | $0.047 \pm 0.017$ | $0.049 \pm 0.218$ | $0.049 \pm 0.154$ | $0.046 \pm 0.020$ | $0.047 \pm 0.220$ |
| DESCN | $0.011 \pm 0.017$ | $0.060 \pm 0.041$ | $0.035 \pm 0.033$ | $0.012 \pm 0.016$ | $0.058 \pm 0.039$ | $0.035 \pm 0.031$ |
| DragonNet | $0.116 \pm 0.079$ | $0.045 \pm 0.022$ | $0.080 \pm 0.195$ | $0.130 \pm 0.279$ | $0.044 \pm 0.024$ | $0.087 \pm 0.207$ |
| CF | $0.014 \pm 0.018$ | $0.051 \pm 0.017$ | $0.033 \pm 0.021$ | $0.016 \pm 0.021$ | $0.052 \pm 0.020$ | $0.034 \pm 0.023$ |
| CFRISW | $0.020 \pm 0.032$ | $0.104 \pm 0.075$ | $0.062 \pm 0.045$ | $0.000 \pm 0.000$ | $0.097 \pm 0.073$ | $0.058 \pm 0.044$ |
| CTRL | $0.070 \pm 0.139$ | $0.043 \pm 0.024$ | $0.057 \pm 0.097$ | $0.071 \pm 0.140$ | $0.041 \pm 0.025$ | $0.056 \pm 0.125$ |
| **TRECA$_+$** | $0.040 \pm 0.067$ | $0.028 \pm 0.025$ | $0.034 \pm 0.032$ | $0.041 \pm 0.068$ | $0.026 \pm 0.022$ | $0.034 \pm 0.032$ |
| **TRECA** | $\mathbf{0.010 \pm 0.015}$ | $\mathbf{0.028 \pm 0.024}$ | $\mathbf{0.019 \pm 0.018}$ | $\mathbf{0.012 \pm 0.017}$ | $\mathbf{0.024 \pm 0.022}$ | $\mathbf{0.018 \pm 0.018}$ |

**Table 4:** Results comparison (mean±std) among baseline and our methods under model misspecificity on real-world datasets. Superscript $M$ means the model is trained with misspecified propensity $\hat{\pi}$.

| | TRECA$_+$ | TRECA$_+^M$ | TRECA | TRECA$^M$ | CFRNet |
| --- | --- | --- | --- | --- | --- |
| Jobs_mono | $0.034 \pm 0.032$ | $0.025 \pm 0.019$ | $0.018 \pm 0.018$ | $0.043 \pm 0.027$ | $0.044 \pm 0.053$ |
| Twins_mono | $0.014 \pm 0.007$ | $0.008 \pm 0.017$ | $0.005 \pm 0.003$ | $0.021 \pm 0.011$ | $0.023 \pm 0.072$ |

second study aims at validating the double robustness of our methods when either model $\hat{\mu}$ or $\hat{\pi}$ is misspecified. We set $\hat{\pi}(x) \equiv 0.25$ and apply our method to classify the treatment responders. Table 4 shows that our methods trained under misspecified model still induce smaller f-regret comparing with baseline method. To check the impact of abstention rate on model performances, in the third study, we shift the abstention rate $\delta$ and compare the performances of our methods with the baseline. Figure 2 manifests that under multiple choices of abstention rates ranging from $0.1$ to $0.5$, both **TRECA** and **TRECA$_+$** generally performs better in learning the classification rule on retained samples comparing with baseline methods at varying abstention rates.

## 5 DISCUSSION

This paper proposes a theoretically-grounded framework to classify treatment responder accurately on retained samples under constraint on abstention rate. Meanwhile, another interesting problem is to consider the dual form which abstain samples as few as possible under limit on misclassification loss, i.e., $\min_r \mathbb{E}[r(X)]$ s.t. $\tilde{L}_\theta(f, r) \leq \varepsilon$. Given $f^*$, the optimal abstention rule $r^*$ that minimizes $E[r(X)]$ is the rule that abstain samples with the highest uncertainty until $\tilde{L}_\theta(f, r) \leq \varepsilon$ for retained samples. This yields a thresholding rule $r^*(x) = I(V_f(x) > \beta')$, which is of the same form as (3). Therefore, our method can also adapt to this dual problem by choosing a different threshold such that the loss constraint on retained samples is satisfied.

ETHICS STATEMENT

We have read and adhere to the Code of Ethics throughout the research. The study does not involve human subjects, practices to dataset releases, potentially harmful insights, potential conflicts of interest of sponsorship, discrimination/bias/fairness concerns, provacy and security issues, legal compliance or research integrity issues.

REPRODUCIBILITY STATEMENT

We have made substantial efforts to ensure the reproducibility of our research. The details of our novel model/algorithm can be found in Section 3 of the main paper with detailed elaboration and proofs provided in the appendix. A link to the anonymous downloadable source code, along with the specification of all dependencies, is provided in anonymous link `https://anonymous.4open.science/r/TRECA-4A0F`. For the datasets used in our experiments, a comprehensive description of the experimental setups, datasets, evaluation metrics and baseline models can be found in Section 4 of the main text.

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

APPENDIX ON TREATMENT RESPONDER CLASSIFICATION WITH ABSTENTION

## A  STATEMENT ON THE USE OF LLM

During the preparation of this manuscript, LLMs were not used for generating research ideas, performing analyses, producing results, or interpreting findings. The authors take full responsibility for all scientific content presented in this work.

## B  ADDITIONAL EXAMPLE AND RELATED WORK

### B.1  ADDITIONAL EXAMPLE ON ABSTENTION LEARNING

**Speed and cost balance in public policy.** The balance between speed and cost has been well addressed as an importance concern in public policy (Vlasova & Rakhmeeva, 2020). For example, during emerging pandemics, practitioners want to decide the area in which public protective policies will be implemented. In this case, treatment is whether to implement the policy in a town or community, and outcome is whether the transmission of pandemic is suppressed locally. To minimize the social impact, decision-makers want to implement public policy only if it is deterministically required. While it is much likely unnecessary to implement protective policy in remote towns with little population and obviously mandatory in metropolis in outbreak regions, there are areas where decision is highly uncertain. For this locations, abstention allows decision-maker to suspend making decision in seek of gathering more data or making more detailed assessment. Treatment responder classification in this scenario saves public welfare on costly restrictions (Vandepitte et al., 2021; Güner et al., 2021), provides a mechanism to suspend decisions for a small proportion of communities, redirecting resources toward closer evaluation on the pandemic prevalence in these areas.

### B.2  RELATED WORK

**CATE Estimation.** CATE refers to the average treatment effect on outcomes for certain subgroup of population characterized by covariates. The most classical methods estimating CATE are based on re-weighting (Austin, 2011; Imai & Ratkovic, 2014; Fong et al., 2018). Other widely used techniques incorporate machine learning, including Causal Forests (CF) (Athey & Imbens, 2016; Wager & Athey, 2018), CEVAE (Louizos et al., 2017), etc. Methods like T-Learner and X-Learner (Künzel et al., 2019) utilize the idea of meta-learning to combine supervised learning and regression methods to enhance robustness and performance. Featured by balancing the distribution between treated and control groups, counterfactual representation learning is capable of producing precise CATE estimation with generalization guarantee, thus a popular method to estimate CATE for complex data (Johansson et al., 2016; Shalit et al., 2017; Shi et al., 2019). There have been a series of advanced CFR-based methods recently, including the combination of importance weighting (Hassanpour & Greiner, 2019), representation decomposition (Wu et al., 2022), optimal transport (Wang et al., 2023) and so on.

**Learning Individual Treatment Rules.** Treatment responder classification can be viewed as a task within learning individual treatment rules (ITR) or treatment regimes. ITR learning is featured by two types of approaches. The first approach is to perform an **estimation-then-decision** procedure which learns the decision rule based on statistical or machine-learning models on heterogeneous treatment effects, such as parametric or semi-parametric regression (Cai et al., 2010) and causal machine learning (Kreif & DiazOrdaz, 2019; Ngufor et al., 2023). The treatments are allocated through analyzing the decision curve on treatment effects in terms of individual biomarkers (Vickers & Elkin, 2006; Fitzgerald et al., 2015), and the uncertainty of the learned policy can be drawn by constructing simultaneous confidence band on the decision curve (Zhou & Ma, 2012; Ma & Zhou, 2017; Guo et al., 2021). The second approach is to treat the problem as a **policy learning** task, directly optimizing the decision rule through regret minimization or reward maximization, which is typically equivalent to a weighted classification problem (Zhao et al., 2012). Some methods separated the treatment effect from the reward, and considered linear, kernel-based Huang & Fong

(2014) or tree based Zhu et al. (2018) methods to discover the decision boundary of the treatment rule. Moreover, taking the robustness into account, Pan & Zhao (2021) proposed a doubly robust estimator to learn the individual treatment rule, and Mo et al. (2021) proposed a distributionally robust method to deal with the situation with training and testing data are not identical. Our work lies in this type of approach by dealing with the binary misclassification loss in selecting treatment responders. Under this scenario, the conditional misclassification risk involves joint distribution of potential outcomes, inducing extra justification on identification and construction of doubly robust estimators.

**Abstention Learning.** Abstention learning allows models to abstain from making decisions on challenging instances to avoid costly errors. The roots of abstention learning can be traced back to Chow (1957; 1970), which introduced the fundamental trade-off between misclassification error and rejection rate in binary classification. Abstention learning framework generally consists of a predictor which predicts the outcome label, and a rejector choosing which samples to abstain Cortes et al. (2016); Mao et al. (2024). Many of the existing works focus on the confidence-based rejection rule, which abstains when the minimum deviation on misclassification probability exceeds a predefined confidence threshold (Herbei & Wegkamp, 2006; Denis & Hebiri, 2020; Zhu & Nowak, 2022). A challenge in cost-based abstention learning is that there is a lack of criterion to determine the cost of abstention. Moreover, since the optimal rule generally involves confidence-cost comparison (Cortes et al., 2024), it is difficult to scale cost and confidence in the same magnitude. In this paper, we formulate the problem by making constraint on the expected proportion of samples being abstained. This method avoids defining an artificial cost, enhancing interpretation of the method. Besides, deferral system or deferral policy learning deals with a similar scenario as abstention learning, in which uncertain decisions are delegated to a secondary agent which is typically a human (Gao & Yin, 2025). Palomba et al. (2025) introduces a causal framework on the deferral system, with a similar goal of minimizing risk under constraint on the proportion of deferred / abstained samples. However, a key difference is that our work focuses on treatment responder selection which involves further discussion on identification and development on robust methods, while Gao & Yin (2025); Palomba et al. (2025) aim at learning the causal effect under RD design or outcome reward maximization.

## C KEY NOTATIONS

**Table 5:** Summary of Notations.

| Notation | Meaning |
|---|---|
| $\min, \max$ | Minimum / Maximum operator. |
| $O_p(\cdot)$ | Probabilistic order. |
| $o_p(\cdot)$ | Converge in probability. |
| $\mathcal{F}$ | Hypothesis class for the predictor $f$. |
| $\mathcal{R}$ | Hypothesis class for the rejector $r$. |
| $f(x)$ | Predictor determining whether an individual is a treatment responder. |
| $r(x)$ | Rejector (abstention rule); $r(x) = 1$ indicates abstention. |
| $T$ | Binary treatment assignment. |
| $Y$ | Observed outcome. |
| $Y(1), Y(-1)$ | Potential outcomes under treatment and control. |
| $R$ | Responder indicator. |
| $\tau(x)$ | Conditional Average Treatment Effect (CATE). |
| $\rho(x)$ | Conditional responding probability. |
| $\pi(x)$ | Propensity score. |
| $V_f(x)$ | Conditional risk. |
| $\beta^*$ | Optimal abstention threshold (CVaR-based). |
| $(u)^- = \min(u, 0)$ | Negative part operator. |
| $I(\cdot)$ | Indicator function. |
| $F_{V_f}$ | Cumulative distribution function of conditional risk. |
| $\hat{\tau}, \hat{\mu}, \hat{\pi}$ | Estimated CATE, outcome regression, and propensity score. |
| $\delta$ | Abstention rate. |
| $\theta$ | Weighting parameter on misclassification types. |

# D SUPPLEMENTARY PROOFS AND THEOREMS

## D.1 PROOF ON PROPOSITION 1

Proposition 1 is a direct result from Theorem 2 in Rockafellar et al. (2000).

## D.2 PROOF ON PROPOSITION 2

Recall that $\rho(X) = \mathbb{P}(R = 1 \mid X = x)$, the proposition is an immediate result from

$$
\begin{aligned}
V_f(x) =& \theta \cdot \mathbb{E}[I(f(X) = +1)I(R = -1) \mid X = x] + (1 - \theta) \cdot \mathbb{E}[I(f(X) = -1)I(R = +1) \mid X = x] \\
=& \frac{\theta}{2}(1 + f(x))(1 - \rho(x)) + \frac{1 - \theta}{2}(1 - f(x))\rho(x) \\
=& \frac{1}{2}f(x)(\theta - \rho(x)) + \frac{1}{2}(\theta + (1 - 2\theta)\rho(x))
\end{aligned}
$$

and

$$
\begin{aligned}
\tau(x) =& \mathbb{E}(Y(1) - Y(-1) \mid X = x) \\
=& 2\mathbb{P}(Y(-1) = -1, Y(+1) = +1 \mid X = x) \\
& - 2\mathbb{P}(Y(-1) = +1, Y(+1) = -1 \mid X = x) \\
=& 2\rho(X = x) - 2\mathbb{P}(Y(-1) < Y(+1) \mid X = x).
\end{aligned}
$$

## D.3 PROOF ON PROPOSITION 3

From Proposition 2, we can rewrite

$$
\begin{aligned}
V_f(x) =& \frac{1}{2}f(x)(\theta - \rho(x)) + \frac{1}{2}(\theta + (1 - 2\theta)\rho(x)) \\
=& \frac{1 - 2\theta - f(x)}{2}\rho(x) + \frac{\theta}{2}(f(x) + 1).
\end{aligned}
$$

Since $f(x) \in \{-1, 1\}$ and $\theta \in [0, 1]$, $1 - 2\theta - f(x) \geq 0$ if $f(x) = -1$, and $1 - 2\theta - f(x) \leq 0$ if $f(x) = 1$. Therefore,

$$
V_f(x) \leq \begin{cases} \frac{1}{2}(-1) \cdot (\theta - \rho^U(x)) + \frac{1}{2}(\theta + (1 - 2\theta)\rho^U(x)) = (1 - \theta)\rho^U(x), & f(x) = -1; \\ \frac{1}{2}(\theta - \rho^L(x)) + \frac{1}{2}(\theta + (1 - 2\theta)\rho^L(x)) = \theta(1 - \rho^L(x)), & f(x) = 1, \end{cases}
$$

which proofs Proposition 3.

## D.4 PROOF ON THEOREM 1

Note that

$$
\begin{aligned}
& \tilde{L}_\theta(f, r) \\
=& \theta\mathbb{P}(f(X) = 1, R = -1, r(X) = 0) + (1 - \theta)\mathbb{P}(f(X) = -1, R = 1, r(X) = 0) \\
=& \mathbb{E}\left[\theta I(f(X) = 1, R = -1) + (1 - \theta)I(f(X) = -1, R = 1) \mid r(X) = 0\right]\mathbb{P}(r(X) = 0) \\
=& \int_{x \in \mathcal{X}} \mathbb{E}\left[\theta I(f(X) = 1, R = -1) + (1 - \theta)I(f(X) = -1, R = 1) \mid X = x\right] \times \\
& p(x \mid r(X) = 0)\mathbb{P}(r(X) = 0)dx \\
=& \int_{x \in \mathcal{X}} V_f(x)p(x \mid r(X) = 0)\mathbb{P}(r(X) = 0)dx,
\end{aligned}
$$

which is the integral on conditional probability density function $p(x \mid r(X) = 0)$ within retained samples weighted by $V_f(x)$. It is straightforward that under constraint $E[r(X)] \leq \delta$, such weighted

integral is minimized at $r^*(X) = I(V_f(X) > F_{V_f}^{-1}(1-\delta))$. For $r = r^*$, we have

$$\tilde{L}_\theta(f, r) = L_{\theta,\delta}(f)$$

$$= \int_{x \in \mathcal{X}} \mathbb{E}\Big\{ \theta I(f(X) = 1, R = -1) + (1-\theta)I(f(X) = -1, R = 1) \mid X = x, V_f(x) \le F_{V_f}^{-1}(1-\alpha) \Big\} \times$$

$$p(x \mid V_f(x) \le F_{V_f}^{-1}(1-\alpha)) \, dx$$

$$= \int_{x \in \mathcal{X}} \mathbb{E}\Big\{ \theta I(f(X) = 1, R = -1) + (1-\theta)I(f(X) = -1, R = 1) \mid X = x \Big\} \times$$

$$p(x \mid V_f(x) \le F_{V_f}^{-1}(1-\alpha)) \, dx$$

$$= \int_{x \in \mathcal{X}} V_f(x) \cdot p(x \mid V_f(x) \le F_{V_f}^{-1}(1-\alpha)) \, dx$$

$$= \mathbb{E}\Big\{ V_f(x) \mid V_f(x) \le F_{V_f}^{-1}(1-\alpha) \Big\}.$$

where $p(x \mid \cdot)$ is the conditional probability density function.

### D.5 PROOF ON THEOREM 2

The Proof on Theorem 2 is a modification from the proofs in Kallus (2023) by adopting different target functions and scrutinizing the differences in objects that CVaR is taken upon. In the following proof, denote $\hat{\mathbb{E}}$ as the sample average. Recall that $(\tau, \mu, \pi)$ are the true values, and $\beta^* = \arg\max_{\beta \in \mathbb{R}} \mathbb{E}l_{\theta,\delta}(f; \tau, \mu, \pi, \beta)$. Note that the following expansion holds

$$\hat{L}_{\theta,\delta}(f; \hat{\tau}, \hat{\mu}, \hat{\pi}) - L_{\theta,\delta}(f)$$

$$= \hat{\mathbb{E}}l_{\theta,\delta}(X, T, Y, f; \hat{\tau}, \hat{\mu}, \hat{\pi}, \hat{\beta}) - \mathbb{E}l_{\theta,\delta}(X, T, Y, f; \tau, \mu, \pi, \beta^*)$$

$$= \mathbb{E}(l_{\theta,\delta}(X, T, Y, f; \hat{\tau}, \hat{\mu}, \hat{\pi}, \hat{\beta}) - l_{\theta,\delta}(X, T, Y, f; \tau, \mu, \pi, \beta^*)) \tag{8}$$

$$+ (\hat{\mathbb{E}} - \mathbb{E})(l_{\theta,\delta}(X, T, Y, f; \hat{\tau}, \hat{\mu}, \hat{\pi}, \hat{\beta}) - l_{\theta,\delta}(X, T, Y, f; \tau, \mu, \pi, \hat{\beta})) \tag{9}$$

$$+ (\hat{\mathbb{E}} - \mathbb{E})(l_{\theta,\delta}(X, T, Y, f; \tau, \mu, \pi, \hat{\beta}) - l_{\theta,\delta}(X, T, Y, f; \tau, \mu, \pi, \beta^*)), \tag{10}$$

It suffices to bound each term separately. To start with, observe that

$$\mathbb{E}l_{\theta,\delta}(X, T, Y, f; \hat{\tau}, \hat{\mu}, \hat{\pi}, \hat{\beta})$$

$$= \hat{\beta} + \frac{1}{1-\delta}\mathbb{E}I(V_f(X; \hat{\tau}) \le \hat{\beta})\Big[ \frac{1}{2}(f(X)+1)\theta + \Big\{ \frac{1}{4} - \frac{\theta}{2} - \frac{1}{4}f(X) \Big\} \times$$

$$\Big\{ \hat{\mu}_1(X) - \hat{\mu}_{-1}(X) + \frac{\pi(X)}{\hat{\pi}(X)}(\mu_1(X) - \hat{\mu}_1(X)) - \frac{1-\pi(X)}{1-\hat{\pi}(X)}(\mu_{-1}(X) - \hat{\mu}_{-1}(X)) \Big\} - \hat{\beta} \Big].$$

We write $A_f(X) = \frac{1}{2}(f(X)+1)\theta$ and $B_f(X) = \frac{1}{4} - \frac{\theta}{2} - \frac{1}{4}f(X)$ in short. For (8), note that

$$|\mathbb{E}(l_{\theta,\delta}(X, T, Y, f; \hat{\tau}, \hat{\mu}, \hat{\pi}, \hat{\beta}) - l_{\theta,\delta}(X, T, Y, f; \tau, \mu, \pi, \hat{\beta}))|$$

$$\le |\mathbb{E}(l_{\theta,\delta}(X, T, Y, f; \hat{\tau}, \hat{\mu}, \hat{\pi}, \hat{\beta}) - l_{\theta,\delta}(X, T, Y, f; \hat{\tau}, \hat{\mu}, \pi, \hat{\beta}))|$$

$$+ |\mathbb{E}(l_{\theta,\delta}(X, T, Y, f; \hat{\tau}, \hat{\mu}, \pi, \hat{\beta}) - l_{\theta,\delta}(X, T, Y, f; \hat{\tau}, \mu, \pi, \hat{\beta}))| \tag{11}$$

$$+ |\mathbb{E}(l_{\theta,\delta}(X, T, Y, f; \hat{\tau}, \mu, \pi, \hat{\beta}) - l_{\theta,\delta}(X, T, Y, f; \tau, \mu, \pi, \hat{\beta}))|$$

$$+ |\mathbb{E}(l_{\theta,\delta}(X, T, Y, f; \tau, \mu, \pi, \hat{\beta}) - l_{\theta,\delta}(X, T, Y, f; \tau, \mu, \pi, \beta))|.$$

We now bound each term in (11) sequentially. For the first term, note that $|B_f(X)| \leq \frac{1-\theta}{2}$,

$$|\mathbb{E}(l_{\theta,\delta}(X,T,Y,f;\hat{\tau},\hat{\mu},\hat{\pi},\hat{\beta}) - l_{\theta,\delta}(X,T,Y,f;\hat{\tau},\hat{\mu},\pi,\hat{\beta}))|$$

$$\leq \frac{1}{1-\delta}\mathbb{E}\left[I[V_f(X;\hat{\tau}) \leq \hat{\beta}]\frac{B_f(X)}{\hat{\pi}(X)}|\pi(X) - \hat{\pi}(X)||\mu_1(X) - \hat{\mu}_1(X)|\right]$$

$$+ \frac{1}{1-\delta}\mathbb{E}\left[I[V_f(X;\tau) \leq \hat{\beta}]\frac{B_f(X)}{1-\hat{\pi}(X)}|\pi(X) - \hat{\pi}(X)||\mu_0(X) - \hat{\mu}_0(X)|\right]$$

$$\leq \frac{1-\theta}{2(1-\delta)\underline{\pi}}\|\pi - \hat{\pi}\|_2 \left(\|\mu_1 - \hat{\mu}_1\|_2 + \|\mu_0 - \hat{\mu}_0\|_2\right).$$

Meanwhile, note that the second term equals zero. For the third term, note that

$$\|V_f(X;\tau) - V_f(X;\hat{\tau})\|_q \leq \frac{1-\theta}{2}\|\tau - \hat{\tau}\|_q.$$

Set $t = \|V_f(X;\tau) - V_f(X;\hat{\tau})\|_q^{\frac{q}{1+q}}$. From the condition that $V_f(X;\tau) - \hat{\beta}$ has density bounded by $c$, applying Hölder's inequality we have

$$|\mathbb{E}(l_{\theta,\delta}(X,T,Y,f;\hat{\tau},\mu,\pi,\hat{\beta}) - l_{\theta,\delta}(X,T,Y,f;\tau,\mu,\pi,\hat{\beta}))|$$

$$= \frac{1}{1-\delta}\left|\mathbb{E}\left[(I[V_f(X;\hat{\tau}) \leq \hat{\beta}] - I[V_f(X;\tau) \leq \hat{\beta}])(V_f(X;\tau) - \hat{\beta})\right]\right|$$

$$\leq \frac{1}{1-\delta}\mathbb{E}\left[|I[V_f(X;\hat{\tau}) \leq \hat{\beta}] \neq I[V_f(X;\tau) \leq \hat{\beta}]||V_f(X;\tau) - \hat{\beta}|\right]$$

$$\leq \frac{1}{1-\delta}\mathbb{E}[|V_f(X;\tau) - V_f(X;\hat{\tau})|I[|V_f(X;\tau) - \hat{\beta}| \leq t]]$$

$$+ \frac{1}{1-\delta}\mathbb{E}[|V_f(X;\tau) - V_f(X;\hat{\tau})|I[|V_f(X;\tau) - V_f(X;\hat{\tau})| > t]]$$

$$\leq \frac{1}{1-\delta}\|V_f(X;\tau) - V_f(X;\hat{\tau})\|_q\mathbb{P}(|V_f(X;\tau) - \hat{\beta}| \leq t)^{(q-1)/q}$$

$$+ \frac{1}{1-\delta}\|V_f(X;\tau) - V_f(X;\hat{\tau})\|_q\mathbb{P}(|V_f(X;\tau) - V_f(X;\hat{\tau})| > t)^{(q-1)/q}$$

$$\leq \frac{1}{1-\delta}\|V_f(X;\tau) - V_f(X;\hat{\tau})\|_q(2ct)^{(q-1)/q} + \frac{1}{1-\delta}\|V_f(X;\tau) - V_f(X;\hat{\tau})\|_q t^{1-q}$$

$$= \frac{1}{1-\delta}((2c)^{(q-1)/q} + 1)\|V_f(X;\tau) - V_f(X;\hat{\tau})\|_q^{\frac{2q}{1+q}}$$

$$\leq \frac{1}{1-\delta}((2c)^{(q-1)/q} + 1)(\frac{1-\theta}{2})^{\frac{2q}{1+q}}\|\tau - \hat{\tau}\|_q^{\frac{2q}{q+1}}.$$

For the fourth term, let $g(\beta) = \mathbb{E}l_{\theta,\delta}(X,T,Y,f;\tau,\mu,\pi,\beta)$. From (11), it is straightforward to check that $g''(\beta) \leq \frac{c}{1-\delta}$. Therefore, from Taylor's expansion we have

$$|\mathbb{E}(l_{\theta,\delta}(X,T,Y,f;\tau,\mu,\pi,\hat{\beta}) - l_{\theta,\delta}(X,T,Y,f;\tau,\mu,\pi,\beta^*))| \leq \frac{c}{2(1-\delta)}(\hat{\beta} - \beta^*)^2.$$

Combining the terms above, we have there exists constants $c_1, c_2, c_3 > 0$ such that

$$|\mathbb{E}(l_{\theta,\delta}(X,T,Y,f;\hat{\tau},\hat{\mu},\hat{\pi},\hat{\beta}) - l_{\theta,\delta}(X,T,Y,f;\tau,\mu,\pi,\hat{\beta}))|$$

$$\leq c_1\|\pi - \hat{\pi}\|_2\|\mu - \hat{\mu}\|_2 + c_2\|\tau - \hat{\tau}\|_q^{\frac{2q}{q+1}} + c_3(\hat{\beta} - \beta^*)^2 \tag{12}$$

From Lemma 1 and Lemma EC.2 in Kallus (2023),

$$|\hat{\beta} - \beta^*| = O_p(\|V_f(X;\tau) - V_f(X;\hat{\tau})\|_q^{\frac{q}{q+1}} + n^{-1/2}) = O_p(\|\hat{\tau} - \tau\|_q^{\frac{q}{q+1}} + n^{-1/2}) = O_p(\epsilon_n^{1/2} \vee n^{-1/2}).$$

Therefore, $(\hat{\beta} - \beta^*)^2 = O_p(\epsilon_n \vee n^{-1})$. Combining the convergence rates on nuisance parameters in Theorem 2, we have (8) is $o_p(\epsilon_n \vee n^{-1/2})$.

Similarly, following the same technic bounding (EC.15) and (EC.16) in Kallus (2023), we have (9), (10) are both $o_p(n^{-1/2})$. Combining Equation (8)-(10) we have $\hat{L}_{\theta,\delta}(f;\hat{\tau},\hat{\mu},\hat{\pi}) - L_{\theta,\delta}(f) = O_p(\epsilon_n \vee n^{-1/2})$.

### D.6 PROOF ON THEOREM 4

Let $\hat{\beta}_U = \arg\max_{\beta \in \mathbb{R}} \mathbb{E} m_{\theta,\delta}(f; \hat{\tau}, \hat{\mu}, \hat{\pi}, \beta)$ and note that $\beta^* = \arg\max_{\beta \in \mathbb{R}} \mathbb{E} I[V_f(X) \leq \beta](V_f(X) - \beta)$. From Assumption 4 there exists $c > 0$ such that the probability density function of $V_f(X)$ is bounded by $c$. Let $c_\rho$ be the upper bound on probability density function of $\rho(X)$ stated in condition *(a)*. Take $t = \|V_f(X) - U_f(X; \hat{\tau})\|_q^{q/(q+1)}$ and observe that

$$\|V_f(X) - U_f(X; \hat{\tau})\|_q \leq \left\| \frac{1 - f(X) - 2\theta}{2} \cdot \max\{\rho(X) - \hat{\rho}^L(X), \hat{\rho}^U(X) - \rho(X)\} \right\|_q \leq (1 - \theta) d_\infty.$$

Hence for any fixed $\beta \in \mathbb{R}$ and $f \in \mathcal{F}$, we have

$$
\begin{aligned}
&\mathbb{P}(I[V_f(X) \leq \beta] \neq I[U_f(X; \hat{\tau}) \leq \beta]) \\
\leq & \mathbb{P}(|V_f(X) - \beta| \leq t) + \mathbb{P}(I[U_f(X; \hat{\tau}) \leq \beta] \neq I[V_f(X) \leq \beta], |V_f(X) - \beta| > t) \\
\leq & 2tc + \mathbb{P}(|V_f(X) - U_f(X; \hat{\tau})| > t) \\
\leq & 2tc + \|V_f(X) - U_f(X; \hat{\tau})\|_q^q t^{-q} \\
= & (2c + 1)\|V_f(X) - U_f(X; \hat{\tau})\|_q^{\frac{q}{q+1}}. \\
\leq & (2c + 1)\|V_f(X) - U_f(X; \hat{\tau})\|_q^{\frac{2q}{q+1}} \leq (2c + 1)((1 - \theta) d_\infty)^{\frac{2q}{q+1}}.
\end{aligned}
\tag{13}
$$

Moreover, from Lemma EC.2 in Kallus (2023),

$$|\hat{\beta}_U - \beta^*| \leq O_p(\|V_f(X) - U_f(X; \hat{\tau})\|_q^{\frac{q}{q+1}} + n^{-1/2}) = O_p((d_\infty)^{\frac{q}{q+1}} + n^{-1/2}).$$

Therefore,

$$
\begin{aligned}
&\mathbb{P}(I[V_f(X) \leq \beta^*] \neq I[V_f(X) \leq \hat{\beta}_U]) \\
\leq & \mathbb{P}(|V_f(X) - \beta^*| \leq |\hat{\beta}_U - \beta^*|) \\
\leq & 2c|\hat{\beta}_U - \beta^*| = O((d_\infty)^{\frac{q}{q+1}} + n^{-1/2}).
\end{aligned}
\tag{14}
$$

Combining (13) with $\beta = \hat{\beta}_U$ and (14), we have

$$
\begin{aligned}
&\mathbb{P}(I[V_f(X) \leq \beta^*] \neq I[U_f(X; \hat{\tau}) \leq \hat{\beta}_U]) \\
\leq & \mathbb{P}(I[V_f(X) \leq \hat{\beta}_U] \neq I[U_f(X; \hat{\tau}) \leq \hat{\beta}_U]) + \mathbb{P}(I[V_f(X) \leq \beta^*] \neq I[V_f(X) \leq \hat{\beta}_U]) \\
= & O((d_\infty)^{\frac{q}{q+1}} \vee n^{-1/2}).
\end{aligned}
\tag{15}
$$

From the expression of $V_f(X)$ in Proposition 2, $f^*(x) = 2I[\rho(x) > \theta] - 1$,[4] and from condition *(b)* $\hat{f}_M(x) = -1$ if $\rho^U(x; \hat{\tau}) \leq \theta$, $\hat{f}_M(x) = 1$ if $\rho^L(x; \hat{\tau}) > \theta$. Consequently,

$$
\begin{aligned}
&\mathbb{P}(f^*(X) \neq \hat{f}_M(X)) \\
\leq & \mathbb{P}(\rho(x) \leq \theta, \hat{\rho}^L(x) > \theta) + \mathbb{P}(\rho(x) > \theta, \hat{\rho}^U(x) \leq \theta) + \mathbb{P}(\hat{\rho}^L(x) < \theta < \hat{\rho}^U(x)) \\
\leq & \mathbb{P}(|\rho(x) - \theta| < d_\infty) + d_\infty \leq (c_\rho + 1) d_\infty,
\end{aligned}
$$

where $c_\rho$ is the upper bound on the density of $\rho(X)$. Hence

$$\mathbb{P}(I[V_{f^*}(X) \leq \beta^*] \neq I[V_{\hat{f}_M}(X) \leq \beta^*]) \leq \mathbb{P}(f^*(X) \neq \hat{f}_M(X)) \leq (c_\rho + 1) d_\infty. \tag{16}$$

Therefore combining (15) and (16), we finally get

$$
\begin{aligned}
&\mathbb{P}(r^*(X) \neq \hat{r}_M(X)) \\
= & \mathbb{P}(I[V_{f^*}(X) \leq \beta^*] \neq I[U_{\hat{f}_M}(X; \hat{\tau}) \leq \hat{\beta}_U]) \\
\leq & \mathbb{P}(I[V_{f^*}(X) \leq \beta^*] \neq I[V_{\hat{f}_M}(X) \leq \beta^*]) \\
& + \mathbb{P}(I[V_{\hat{f}_M}(X) \leq \hat{\beta}_U] \neq I[U_{\hat{f}_M}(X; \hat{\tau}) \leq \hat{\beta}_U]) + \mathbb{P}(I[V_{\hat{f}_M}(X) \leq \beta^*] \neq I[V_{\hat{f}_M}(X) \leq \hat{\beta}_U]) \\
\leq & O(d_\infty) + O((d_\infty)^{\frac{q}{q+1}} \vee n^{-1/2}) = O(d_\infty \vee n^{-1/2}),
\end{aligned}
$$

which completes the proof.

---

[4]To be rigorous, $f^* \in \mathcal{F}^{opt}$, where $\mathcal{F}^{opt}$ is the set of predictors at minimization of $L_{\theta,\delta}(f)$.

### D.7 PROOF ON THEOREM 3

We proceed similar as in Kallus (2023)by checking the orthogonality of parameters $(\tau, \mu, \pi, \beta)$ ensuing double robustness. Note that

$$
\mathbb{E} m_{\theta,\delta}(X, T, Y, f; \hat{\tau}, \hat{\mu}, \hat{\pi}, \hat{\beta})
$$

$$
= \hat{\beta} + \frac{1}{1-\delta} \mathbb{E} I(U_f(X; \hat{\tau}) \leq \hat{\beta}) \Big[ U_f(X; \hat{\tau}) - \hat{\beta}
$$

$$
+ \partial_\tau U_f(X; \hat{\tau}) \Big\{ \frac{\pi(X)}{\hat{\pi}(X)} (\mu_1(X) - \hat{\mu}_1(X)) - \frac{1 - \pi(X)}{1 - \hat{\pi}(X)} (\mu_{-1}(X) - \hat{\mu}_{-1}(X)) \Big\} \Big].
$$

The double robustness in $e, \mu$ comes from the observation that $\mathbb{E} m_{\theta,\delta}(X, T, Y, f; \hat{\tau}, \mu, \pi, \hat{\beta})$ is a combination of CATE weighted by fixed weight function in $X$ given $\hat{\tau}, \hat{\beta}$. The orthogonality of $\tau, \beta^*$ comes from the fact that let $h(\hat{\tau}, \hat{\beta}) = \mathbb{E} m_{\theta,\delta}(X, T, Y, f; \hat{\tau}, \mu, \pi, \hat{\beta}) = \hat{\beta} + \frac{1}{1-\delta} \mathbb{E} I(U_f(X; \hat{\tau}) \leq \hat{\beta})(U_f(X; \hat{\tau}) - \hat{\beta})$. Then from the same technic bounding the third the fourth term in (11), there exists constants $c_1', c_2' > 0$ such that

$$
|h(\hat{\tau}, \hat{\beta}) - h(\tau, \beta^*)| \leq c_1' \|\hat{\tau} - \tau\|_q^{\frac{2q}{q+1}} + c_2' (\hat{\beta} - \beta^*)^2.
$$

Therefore, $h(\hat{\tau}, \hat{\beta})$ has zero derivative at $(\tau, \beta^*)$ and the orthogonality holds following the same argument after (14) in Kallus (2023). In conclusion, the double robustness and ensuing orthogonality continues to hold for upper bound estimator (6).

### D.8 GENERALIZATION BOUND ON MISCLASSIFICATION LOSS WITH ABSTENTION UNDER MONOTONICITY

We derive the generalization bound on the naive estimator $\hat{L}_{\theta,\delta}^{\text{naive}}(f) = \sup_{\beta \in \mathbb{R}} \{ \beta + \frac{1}{n(1-\delta)} \sum_{i=1}^n (V_f(X_i) - \beta)_- \}$ on loss function $L_{\theta,\delta}(f)$ measuring the complexity function class using Empirical Rademacher Complexity in Definition 1.

**Definition 1 (Empirical Rademacher Complexity)** *Let $\mathcal{F}$ be a family of functions mapping from $\mathcal{X}$ to $\{-1, +1\}$. Then, for a fixed set $S = (x_1, \ldots, x_n)$ of size $n$ sampled from $\mathcal{X}$, the empirical Rademacher complexity of $\mathcal{F}$ with respect to the sample $S$ is*

$$
\mathfrak{R}_n(\mathcal{F}) = \mathbb{E}_\sigma \left[ \sup_{f \in \mathcal{F}} \frac{1}{n} \left| \sum_{i=1}^n \sigma_i f(x_i) \right| \right],
$$

*where $\sigma = (\sigma_1, \ldots, \sigma_n)^\top$, with $\sigma_i$'s be independent identically distributed random variables taking values in $\{-1, +1\}$ with fair probability.*

**Theorem 5 (Generalization Bound on Surrogate Loss)** *Let $\mathfrak{R}_n(\mathcal{H})$ be the empirical Rademacher complexity for a function class $\mathcal{H}$ with respect to the training samples $\mathcal{D}$ with size $n$. Then under Assumption 5, there exists $c = \max_{x \in \mathcal{X}} \frac{1}{2\delta} |\theta - \frac{\tau(x)}{2}|$ such that with probability at least $1 - \epsilon$,*

$$
L_{\theta,\delta}(f) \leq \min_{f \in \mathcal{F}} \hat{L}_{\theta,\delta}^{\text{naive}}(f) + 2c \mathfrak{R}_n(\mathcal{F}) + 5c \sqrt{\frac{2 \log \frac{8}{\epsilon}}{n}}.
$$

### PROOF ON THEOREM 5

Let $\beta^* = \arg \max_{\beta \in \mathbb{R}} g_{\theta,\delta}(f; \beta)$. with $g_{\theta,\delta}(f; \beta) = \beta + \frac{1}{\delta} E(V_f(X) - \beta)_-$. From Theorem 1, under Assumption 5, $V_f(x) = \frac{1}{2} f(x) \{ \theta - \frac{\tau(x)}{2} \} + \frac{1}{2} \{ \theta + (1 - 2\theta) \frac{\tau(x)}{2} \}$ is a Lipschitz function in terms of $f$ with Lipschitz constant $c = \max_{x \in \mathcal{X}} \frac{1}{2\delta} |\theta - \frac{\tau(x)}{2}|$. Therefore, with probability at least $1 - \epsilon$, $\forall \beta \in \mathbb{R}$,

$$
g_{\theta,\delta}(f; \beta) \leq \hat{l}_{\theta,\delta}(f; \beta) + 2c \mathfrak{R}_n(\mathcal{F}) + 5c \sqrt{\frac{2 \log \frac{8}{\epsilon}}{n}}
$$

with $\hat{g}_{\theta,\delta}(f) = \beta + \frac{1}{n(1-\delta)} \sum_{i=1}^{n} (V_f(X_i) - \beta)_-$. Take $\beta = \beta^*$ and recall that $L_{\theta,\delta}(f) = \sup_{\beta \in \mathbb{R}} g_{\theta,\delta}(f;\beta)$, we have

$$L_{\theta,\delta}(f) = g_{\theta,\delta}(f;\beta^*) \leq \hat{g}_{\theta,\delta}(f;\beta^*) + 2c\mathfrak{R}_n(\mathcal{F}) + 5c\sqrt{\frac{2\log\frac{8}{\epsilon}}{n}}$$

$$\leq \sup_{\beta \in \mathbb{R}} \hat{g}_{\theta,\delta}(f;\beta) + 2c\mathfrak{R}_n(\mathcal{F}) + 5c\sqrt{\frac{2\log\frac{8}{\epsilon}}{n}}$$

$$= \hat{L}_{\theta,\delta}^{\text{naive}}(f) + 2c\mathfrak{R}_n(\mathcal{F}) + 5c\sqrt{\frac{2\log\frac{8}{\epsilon}}{n}}.$$

The proof is done.

# E ADDITIONAL EXPERIMENTS AND ALGORITHMS

## E.1 DETAILS ON DATASETS AND PREPROCESSING.

To measure the effectiveness of the proposed methods, we conduct extensive experiments based on two classical real-world datasets, **Twins** (Almond et al., 2005) and **Jobs** (LaLonde, 1986). The Twins dataset seeks to explore the risk of low-birth weight on the mortality of the baby. It consists of 11,984 pairs of twins born in the USA between 1989 and 1991, including 50 covariates for the twin pair such as mother and father age and education, health complications and so on. Following Louizos et al. (2017), treatment $T = 1$ indicates that the baby is heavier than his/her cousin, and the outcome is the mortality for the twins. For each pair of twins, we randomly choose which baby we can observe with equal probability. The Jobs dataset is based on the National Supported Work program (Smith & Todd, 2005), in which the treatment is job training and the outcome is employment status and income (Shalit et al., 2017). The dataset consists of 3212 units (9% treated, 91% control), with 15 covariates measuring the attributes of job seekers. We generate the counterfactual outcomes on Jobs following (Wu et al., 2025), and substitute $Y(1)$ with $Y^{\text{new}}(1) = \max\{Y(1), Y(-1)\}$ to ensure monotonicity (Assumption 5). We denote the transformed datasets as **Twins_mono** and **Jobs_mono**. Since the counterfactual outcomes are known in both datasets, the true treatment responder indicator $R$ is known, enabling regret computation. With the main results evaluated on transformed dataset, we also evaluate the performances of our method on the original dataset when monotonicity violates in ablation studies. Training/validation/testing sets are split at ratio of $63/27/10$ in Twins and $80/10/10$ in Jobs.

## E.2 ADDITIONAL DETAILS ON MODEL SETUP, CONFIGURATIONS AND TRAINING PROCESS

In both **TRECA** and **TRECA$_+$**, we use CFRNet (Shalit et al., 2017) in train the CATE estimator $\hat{\tau}$ and condition expectation $\hat{\mu}$. CFRNet is chosen for two main reasons. (1) Through balancing representation distributions in factual and counterfactual outcomes, experimental results shows that counterfactual representation learning (CFR) has good performance in classifying treatment responders. Previous work also recommends CFR-based methods to learn CATE in treatment responder classification task (Wu et al., 2025). (2) The bifurcated structure of CFRNet with each branch predicting factual and counterfactual outcomes respectively ensures good fitness on $\mu$, which is also theoretically guaranteed in Theorem 1 in (Shalit et al., 2017) to ensure low generalization bound. We choose MLP with two hidden layers to learn the rejector and predictor.

All experiments were conducted on a Core i5-1155G7 Laptop CPU. The batch size is set as 300, and training iteration set as 1000 for all models. The optimal learning rate is set as 0.0005 for TRECA. We employ the Adaptive Moment Estimation (Adam) optimizer during model training. In the TRECA model, the representation network consists of two layers, the propensity network contains three layers, and the outcome network includes two layers. All hidden layers are set to a dimensionality of 16.

### E.3 ADDITIONAL RESULTS ON TWINS AND JOBS

We also compare our methods with baseline models on the original dataset **Twins** and **Jobs** when monotonicity violates. The result shown in Table 6 indicates the superiority on our methods over most baselines among retained samples. Note that CTRL developed in Wu et al. (2025) also performs well since it is designed to with treatment responder classification without monotonicity assumption. In scenarios when monotonous assumption holds, our methods outperform CTRL as shown in Table 2.

**Table 6:** Results (mean±std) of FPR, FNR, f-regret on Twins/Jobs datasets with abstention rate = 30%

| Method | Within-samples on Twins | | | Out-of-samples on Twins | | |
|---|---|---|---|---|---|---|
| | FPR ↓ | FNR ↓ | f-regret ↓ | FPR ↓ | FNR ↓ | f-regret ↓ |
| T_learner | 0.054 ± 0.187 | 0.009 ± 0.003 | 0.032 ± 0.093 | 0.053 ± 0.185 | 0.007 ± 0.006 | 0.030 ± 0.092 |
| X_learner | 0.048 ± 0.180 | 0.008 ± 0.003 | 0.028 ± 0.089 | 0.048 ± 0.183 | 0.005 ± 0.005 | 0.027 ± 0.091 |
| CFRNet | 0.017 ± 0.091 | **0.001 ± 0.002** | 0.009 ± 0.046 | 0.021 ± 0.088 | 0.007 ± 0.004 | 0.014 ± 0.044 |
| CFRNetv2 | 0.037 ± 0.182 | 0.009 ± 0.003 | 0.023 ± 0.090 | 0.037 ± 0.184 | 0.007 ± 0.005 | 0.022 ± 0.091 |
| CEVAE | 0.043 ± 0.068 | 0.009 ± 0.003 | 0.026 ± 0.033 | 0.053 ± 0.070 | 0.005 ± 0.004 | 0.029 ± 0.035 |
| ESCFR | 0.005 ± 0.036 | 0.002 ± 0.003 | 0.004 ± 0.018 | 0.011 ± 0.035 | 0.006 ± 0.005 | 0.009 ± 0.017 |
| DeRCFR | 0.036 ± 0.167 | 0.001 ± 0.002 | 0.019 ± 0.084 | 0.040 ± 0.170 | 0.006 ± 0.005 | 0.023 ± 0.084 |
| DESCN | 0.100 ± 0.270 | 0.002 ± 0.002 | 0.051 ± 0.135 | 0.104 ± 0.269 | 0.006 ± 0.006 | 0.055 ± 0.134 |
| DragonNet | 0.048 ± 0.195 | 0.002 ± 0.004 | 0.025 ± 0.097 | 0.053 ± 0.195 | 0.006 ± 0.004 | 0.029 ± 0.097 |
| CF | 0.011 ± 0.039 | 0.001 ± 0.003 | 0.006 ± 0.020 | 0.016 ± 0.045 | 0.008 ± 0.004 | 0.012 ± 0.022 |
| CFRISW | 0.000 ± 0.000 | 0.009 ± 0.003 | 0.005 ± 0.001 | 0.000 ± 0.000 | 0.007 ± 0.005 | 0.003 ± 0.003 |
| CTRL | 0.024 ± 0.054 | 0.004 ± 0.004 | 0.014 ± 0.028 | 0.031 ± 0.058 | 0.007 ± 0.005 | 0.019 ± 0.029 |
| **TRECA$_+$** | 0.035 ± 0.031 | 0.009 ± 0.003 | 0.022 ± 0.016 | 0.068 ± 0.092 | 0.007 ± 0.005 | 0.038 ± 0.045 |
| **TRECA** | **0.002 ± 0.004** | 0.010 ± 0.003 | 0.006 ± 0.003 | 0.006 ± 0.011 | 0.007 ± 0.004 | **0.006 ± 0.005** |

| Method | Within-samples on Jobs | | | Out-of-samples on Jobs | | |
|---|---|---|---|---|---|---|
| | FPR ↓ | FNR ↓ | f-regret ↓ | FPR ↓ | FNR ↓ | f-regret ↓ |
| T_learner | 0.167 ± 0.099 | 0.126 ± 0.111 | 0.142 ± 0.056 | 0.164 ± 0.104 | 0.132 ± 0.118 | 0.145 ± 0.062 |
| X_learner | 0.150 ± 0.090 | 0.142 ± 0.118 | 0.145 ± 0.059 | 0.152 ± 0.099 | 0.144 ± 0.126 | 0.147 ± 0.067 |
| CFRNet | 0.109 ± 0.065 | 0.084 ± 0.060 | **0.094 ± 0.037** | 0.121 ± 0.069 | 0.113 ± 0.062 | 0.116 ± 0.035 |
| CEVAE | 0.460 ± 0.097 | **0.004 ± 0.013** | 0.696 ± 0.145 | 0.433 ± 0.116 | **0.003 ± 0.011** | 0.655 ± 0.174 |
| ESCFR | 0.118 ± 0.062 | 0.109 ± 0.090 | 0.340 ± 0.127 | 0.116 ± 0.080 | 0.118 ± 0.096 | 0.351 ± 0.149 |
| DeRCFR | 0.129 ± 0.084 | 0.091 ± 0.043 | 0.330 ± 0.120 | 0.128 ± 0.092 | 0.098 ± 0.044 | 0.339 ± 0.130 |
| DESCN | 0.129 ± 0.088 | 0.122 ± 0.121 | 0.376 ± 0.168 | 0.120 ± 0.093 | 0.126 ± 0.117 | 0.370 ± 0.167 |
| DragonNet | 0.142 ± 0.105 | 0.103 ± 0.104 | 0.368 ± 0.167 | 0.141 ± 0.110 | 0.104 ± 0.110 | 0.368 ± 0.175 |
| CF | 0.114 ± 0.032 | 0.098 ± 0.039 | 0.318 ± 0.083 | 0.113 ± 0.043 | 0.103 ± 0.041 | 0.324 ± 0.091 |
| CFRISW | 0.296 ± 0.184 | 0.123 ± 0.208 | 0.192 ± 0.057 | 0.289 ± 0.185 | 0.125 ± 0.212 | 0.191 ± 0.063 |
| CTRL | 0.169 ± 0.130 | 0.080 ± 0.049 | 0.116 ± 0.041 | 0.160 ± 0.131 | 0.077 ± 0.048 | **0.110 ± 0.042** |
| **TRECA$_+$** | 0.111 ± 0.156 | 0.204 ± 0.182 | 0.158 ± 0.072 | 0.110 ± 0.165 | 0.213 ± 0.187 | 0.161 ± 0.077 |
| **TRECA** | **0.036 ± 0.039** | 0.289 ± 0.168 | 0.163 ± 0.069 | **0.037 ± 0.039** | 0.284 ± 0.157 | 0.161 ± 0.064 |

### E.4 EXPERIMENTS ON NEURAL NETWORK DEPTH

To examine whether impact of the space complexity in hypothesis class of predictors $\mathcal{F}$ on model performance, we conduct an additional set of experiments to evaluate the performance of our method under different layers of predictor networks that correspond to different levels of space complexity. We vary the number of layers in the predictor network and evaluate architectures with 4, 10, 30, and 50 layers under the settings of $\theta = 0.5$ and an abstention rate of $\delta = 0.1$ on the Twins_mono dataset. We also compare the performance with the predictor trained from logistic regression, i.e., $\mathbb{P}(f(x) = 1) = logit(x^\top \alpha)$. The results are summarized in Table 7.

**Table 7:** Performance of $f$ under different layer settings.

| layer of $f$ | 4 | 10 | 30 | 50 | logistic |
|---|---|---|---|---|---|
| **f-regret** | 0.0035 | 0.0035 | 0.0034 | **0.0032** | 0.0547 |
| **f-regret-abs** | 0.3485 | 0.2273 | 0.0758 | 0.0530 | 0.2930 |
| **f-regret-overall** | 0.0388 | 0.0264 | 0.0109 | 0.0031 | 0.1308 |

**Algorithm 1** **T**reatment **RE**sponder **C**lassification with **A**bstention under Monotonicity Assumption (**TRECA**)

**Require:** Level $\alpha \in (0,1)$, $k, m \in \mathbb{N}$, data $\mathcal{D} = \{(X_i, T_i, Y_i) : i = 1, \dots, n\}$, initialized $\hat{\mu}, \hat{\pi}, \hat{\tau}$.
1: Estimate $\hat{\pi}, \hat{\mu}$ through minimizing MSE $\frac{1}{n} \sum_{i=1}^{n} (T_i - \hat{\pi}(X_i))^2$ and $\frac{1}{n} \sum_{i=1}^{n} (Y_i - \hat{\mu}_{T_i}(X_i))^2$.
   Let $\omega_f$ and $\omega_\tau$ be the parameters of $\hat{f}$ and $\hat{\tau}$.
2: **while** not converge **do**
3:     Sample a mini-batch of size $k$ with index $K = \{i_1, \dots, i_k\}$ from $\mathcal{D}$.
4:     $s = 0$.
5:     **for** $s \leq m$ **do**
6:         Update $\hat{\beta} = \hat{\beta} + \epsilon_\beta \frac{1}{k} \sum_{i \in K} \partial_\beta l_{\theta,\delta}(X_i, T_i, Y_i, f; \hat{\tau}, \hat{\mu}, \hat{\pi}, \beta)$.
7:         Update $\omega_\tau = \omega_\tau - \epsilon_\tau \frac{1}{k} \sum_{i \in K} \partial_{\omega_\tau} l_{\theta,\delta}(X_i, T_i, Y_i, f; \hat{\tau}, \hat{\mu}, \hat{\pi}, \hat{\beta})$.
8:         Update $\omega_f = \omega_f - \epsilon_f \frac{1}{k} \sum_{i \in K} \partial_{\omega_f} l_{\theta,\delta}(X_i, T_i, Y_i, f; \hat{\tau}, \hat{\mu}, \hat{\pi}, \hat{\beta})$.
9:         $s = s + 1$.
10:     **end for**
11: **end while**
12: Compute $V_{\hat{f}}(X_i)$ from Proposition 1.
13: Set $\hat{q}_\delta = \inf\{q : \frac{1}{n} \sum_{i=1}^{n} I(V_{\hat{f}}(X_i) \geq q) - \delta \leq 0\}$.
14: Estimate the abstention rule $\hat{r}(X_i) = I(V_{\hat{f}}(X_i, \hat{\tau}) > \hat{q}_\delta)$.

---

**Algorithm 2** **T**reatment **RE**sponder **C**lassification with **A**bstention without Monotonicity Assumption (**TRECA**$_+$)

**Require:** Level $\alpha \in (0,1)$, $k, m \in \mathbb{N}$, data $\mathcal{D} = \{(X_i, T_i, Y_i) : i = 1, \dots, n\}$, initialized $\hat{\mu}, \hat{\pi}, \hat{\tau}$.
1: Estimate $\hat{\pi}, \hat{\mu}$ through minimizing MSE $\frac{1}{n} \sum_{i=1}^{n} (T_i - \hat{\pi}(X_i))^2$ and $\frac{1}{n} \sum_{i=1}^{n} (Y_i - \hat{\mu}_{T_i}(X_i))^2$.
   Let $\omega_f$ and $\omega_\tau$ be the parameters of $\hat{f}$ and $\hat{\tau}$.
2: **while** not converge **do**
3:     Sample a mini-batch of size $k$ with index $K = \{i_1, \dots, i_k\}$ from $\mathcal{D}$.
4:     $s = 0$.
5:     **for** $s \leq m$ **do**
6:         Update $\hat{\beta} = \hat{\beta} + \epsilon_\beta \frac{1}{k} \sum_{i \in K} \partial_\beta m_{\theta,\delta}(X_i, T_i, Y_i, f; \hat{\tau}, \hat{\mu}, \hat{\pi}, \beta)$.
7:         Update $\omega_\tau = \omega_\tau - \epsilon_\tau \frac{1}{k} \sum_{i \in K} \partial_{\omega_\tau} m_{\theta,\delta}(X_i, T_i, Y_i, f; \hat{\tau}, \hat{\mu}, \hat{\pi}, \hat{\beta})$.
8:         Update $\omega_f = \omega_f - \epsilon_f \frac{1}{k} \sum_{i \in K} \partial_{\omega_f} m_{\theta,\delta}(X_i, T_i, Y_i, f; \hat{\tau}, \hat{\mu}, \hat{\pi}, \hat{\beta})$.
9:         $s = s + 1$.
10:     **end for**
11: **end while**
12: Compute $U_{\hat{f}}(X_i)$ from Proposition 1.
13: Set $\hat{q}_\delta = \inf\{q : \frac{1}{n} \sum_{i=1}^{n} I(U_{\hat{f}}(X_i) \geq q) - \delta \leq 0\}$.
14: Estimate the abstention rule $\hat{r}(X_i) = I(U_{\hat{f}}(X_i, \hat{\tau}) > \hat{q}_\delta)$.

---

The results indicate that increasing the depth of the predictor network consistently improves model performance. Specifically, the regret on the retained samples (**f-regret**) exhibits a slight but steady decrease as the depth grows from 4 to 50 layers, together with improvements on the regret on abstained samples (**f-regret-abs**) and the overall sample (**f-regret-overall**). The **f-regret** using neural network is smaller than that using logistic regression. These findings suggest that deeper predictor networks can better capture the uncertainty structure and further enhance the effectiveness of the abstention mechanism.

### E.5 EXPERIMENTS ON TIME AND SPACE COMPLEXITY

To assess and compare the computational efficiency of different models, we conducted experiments measuring their running time under identical conditions. All methods were trained on the Twins_mono dataset using the same experimental settings($\theta = 0.5$, abstention rate $\delta = 0.3$) and

the same hardware configuration to ensure strict comparability. For each method, we recorded the total time required to complete 1000 training epochs. The results are summarized in Table 8.

**Table 8:** Running time and memory comparison across different models.

| Model | Time | Rate |
|---|---|---|
| DragonNet | 4.30s | 10.19% |
| DESCN | 5.18s | 12.28% |
| ESCFR | 8.97s | 21.27% |
| CFRISW | 12.64s | 29.97% |
| **TRECA (ours)** | **42.18s** | **100%** |
| CFRNet | 86.83s | 205.86% |
| DeRCFR | 100.89s | 239.19% |
| CTRL | 134.20s | 318.16% |
| CEVAE | 183.75s | 435.63% |

(a) Running time comparison.

| Model | Memory (standardized) |
|---|---|
| CFRNet | 0.5 |
| DragonNet | 0.7 |
| CFRISW | 0.7 |
| **TRECA (ours)** | **1.0** |
| ESCFR | 1.3 |
| CTRL | 1.4 |
| DESCN | 1.8 |
| DeRCFR | 1.8 |
| CEVAE | 2.5 |

(b) Memory usage comparison.

The results indicate that simpler meta-learning methods such as DragonNet and DESCN achieve the shortest training times, while more complex or deep generative models such as CEVAE and CTRL require substantially longer computation. Taking TRECA as a reference point (100%), we ranked all other methods by their relative running-time ratios. As shown in the table, our proposed model TRECA lies in the middle of the spectrum. Moreover, the models that train faster than TRECA consistently exhibit larger f-regret, indicating that their efficiency comes at the cost of reduced predictive accuracy. In contrast, TRECA maintains moderate computational complexity while achieving strong predictive performance. This demonstrates that TRECA provides a favorable balance between efficiency and accuracy.

For space complexity, we record and compare the system memory training our methods and baseline methods. For standardization, we set the memory for our method as 1. Table 8 shows that our method has a medium level of memory occupation. In particular, the required memory is less than that of CTRL, another treatment responder classification method using joint learning.

### E.6 EXPERIMENTS ON BASELINE MODELS UNDER UNCERTAINTY-BASED ABSTENTION STRATEGY

To compare TRECA and baseline methods under consistent abstention conditions, we conducted additional experiments using a unified uncertainty-based abstention procedure. For baseline estimators on CATE $\hat{\tau}(x)$, following the derivation in Proposition 2, we compute the conditional risk

$$\hat{V}_f(x) = \frac{1}{2}f(x)\{\theta - \hat{\rho}(x)\} + \frac{1}{2}\{\theta + (1 - 2\theta)\hat{\rho}(x)\}$$

with $\hat{\rho}(x) = \hat{\tau}(x)/2$, and abstain the samples with $100\delta\%$ highest conditional risk to evaluate each model solely on the remaining (retained) samples. All models were trained on the `Twins_mono` dataset with the same experimental setting ($\theta = 0.5$, $\delta = 0.3$) and identical hardware configuration to ensure comparability. The models are evaluated through **f-regret**, **f-regret-abs**, and **f-regret-overall**, regrets on retained, abstained and entire samples respectively. The updated results are summarized in Table 9.

The results show that when the same uncertainty-based abstention strategy is used, TRECA still outperforms the baseline method. In particular, TRECA achieves the lowest **f-regret**, indicating that our method more effectively improves performance on the retained samples by appropriately abstaining on uncertain cases. These findings further validate the robustness and effectiveness of TRECA under consistent abstention strategies.

### E.7 EXPERIMENTS ON VIOLATION OF THE UNCONFOUNDEDNESS ASSUMPTION

The unconfoundedness assumption warrants further discussion when abstention is incorporated into causal decision-making. we extend our method to accommodate potential violations of the uncon-

**Table 9:** Comparison of **f-regret** under uncertainty-based abstention Strategy.

| Model | Within-samples | | | Out-of-samples | | |
|---|---|---|---|---|---|---|
| | f-regret | f-regret-abs | f-regret-overall | f-regret | f-regret-abs | f-regret-overall |
| TRECA | **0.003 ± 0.002** | 0.352 ± 0.171 | 0.039 ± 0.017 | **0.003 ± 0.003** | 0.322 ± 0.166 | 0.036 ± 0.017 |
| CTRL | 0.004 ± 0.001 | 0.273 ± 0.029 | 0.085 ± 0.009 | 0.006 ± 0.002 | 0.283 ± 0.040 | 0.089 ± 0.012 |
| DragonNet | 0.084 ± 0.166 | 0.370 ± 0.190 | 0.279 ± 0.207 | 0.087 ± 0.163 | 0.372 ± 0.196 | 0.276 ± 0.211 |
| DeRCFR | 0.098 ± 0.191 | 0.470 ± 0.032 | 0.456 ± 0.114 | 0.103 ± 0.190 | 0.481 ± 0.020 | 0.460 ± 0.117 |
| CFRNet | 0.018 ± 0.030 | 0.063 ± 0.065 | 0.023 ± 0.032 | 0.016 ± 0.026 | 0.065 ± 0.064 | 0.021 ± 0.027 |

**Table 10:** Comparison of f-regret across models when unconfoundedness possibly violates.

| Model | f-regret | |
|---|---|---|
| | Within-samples | Out-of-samples |
| TRECA | $0.003 \pm 0.002$ | $0.003 \pm 0.003$ |
| TRECA(w/o unconf.) | $0.004 \pm 0.002$ | $0.004 \pm 0.002$ |
| CTRL | $0.004 \pm 0.001$ | $0.006 \pm 0.002$ |
| DragonNet | $0.084 \pm 0.166$ | $0.087 \pm 0.163$ |
| DeRCFR | $0.098 \pm 0.191$ | $0.103 \pm 0.190$ |
| CFRNet | $0.018 \pm 0.030$ | $0.016 \pm 0.026$ |

foundedness assumption. In our experiments, we use the `Twins_mono` dataset and set $\theta = 0.5$ and the abstention rate $\delta = 0.1$. We compare our method **TRECA** with its extended version when the unconfoundedness assumption is relatex, marked by **w/o unconf.** The updated results are summarized in Table 10.

The results demonstrate that with slightly increase in regret due to relaxation in unconfoundedness assumptions, TRECA still achieves notably lower regret classifying treatment responders on retained samples.

