# OpenReview forum: "Treatment Responder Classification with Abstention"
_ICLR.cc/2026/Conference — Submitted to ICLR 2026_

### Official Review · Reviewer_YhGp · 2025-10-30

**Soundness:** 3
**Presentation:** 3
**Contribution:** 2
**Rating:** 6
**Confidence:** 3

**Summary:**

The paper proposes an adaptation of abstention methods to treatment responder classification.
The authors present a principled method grounded in theory to correctly estimate the functions of interest and evaluate their approach over two existing datasets.

**Strengths:**

The paper's main strengths:

1. The extension of abstention to the setting of treatment responder classification is very interesting
2. The presented method is theoretically grounded
3. The paper has the potential to be impactful

**Weaknesses:**

The main concerns I have are:

*Soundness*: The proofs rely on the unconfoundedness assumption, which is not properly discussed in my opinion. Indeed, a classical argument supporting abstention (as also acknowledged by the authors e.g., in lines 69-70; ) is the possibility of obtaining extra information on some difficult-to-classify instances. If new variables are involved, I think one might need an ad hoc notion of unconfoundedness.

*Clarity:* The paper notation is very dense, and sometimes it also seems not very consistent. For example, it seems to me that both $\mathbb{E}$ and $E$ are being used for denoting expected values (es within Eq. 1 and Proposition 1, respectively).


*Empirical Evaluation:* I am not sure whether I completely understand how the empirical evaluation was carried out.
In particular, the authors state:

> For fair comparison, we randomly abstain the samples on baselines to ensure the metrics are computed on the same amount of samples.

I am missing the point here. Why not consider some naive "abstention" baselines rather than randomly abstaining? For instance, to better isolate the contribution of TRECA, it would be informative to include some abstention baseline that does not rely on the proposed doubly robust approach. For example, why not abstain when the uncertainty (e.g., using some ensemble for $\tau(x)$ estimation or some variance measure) is too large (i.e, above a certain quantile)? This would clarify whether the observed performance gains stem from the causal abstention risk itself or from the underlying CATE estimation quality.

*Relation to Prior Work:* I think the paper could benefit from adding another related work paragraph mentioning a few recent works that investigated (from different perspectives) abstention and causal inference.
For instance, [1] proposes to consider a *counterfactual score* of an abstaining classifier, defined as the expected performance of the classifier had it not been allowed to abstain.
Moreover, some recent works connect causality and deferral (i.e., rather than simply abstaining, we involve another predictor in the decision): in [2] the authors propose a framework to "causally" evaluate classifiers that defer the decision to humans, while in [3] the authors propose an approach to learn deferral policy robust to confounding.

 I think discussing these works might better position the work in the current literature

**References**


[1] Choe, Yo Joong, Aditya Gangrade, and Aaditya Ramdas. "Counterfactually comparing abstaining classifiers." Advances in Neural Information Processing Systems 36 (2023): 28281-28293.

[2] Palomba, F., Pugnana, A., Alvarez, J. M., & Ruggieri, S. (2025, April). A Causal Framework for Evaluating Deferring Systems. In International Conference on Artificial Intelligence and Statistics (pp. 2143-2151). PMLR.

[3] Gao, R., & Yin, M. (2025, April). Confounding-Robust Deferral Policy Learning. In Proceedings of the AAAI Conference on Artificial Intelligence (Vol. 39, No. 13, pp. 14238-14246).

**Questions:**

I would like the authors to discuss the above weaknesses.

I have a few other remarks I would like the authors to discuss:


*Technical Differences for Novelty* I appreciate the bridging of treatment responder classification and abstention. Still, I think the authors might emphasize better the differences between some of their results and the results from Franc et al., 2023. The main differences I see are some technical concerns related to the estimation due to the nuisance parameters, but in spirit, the optimal abstention strategy still consists of thresholding some specific conditional risk. Is it true or am I missing something?

---

> ### Author Response · Authors · 2025-11-24
>
> Thank you for your general appreciation on the impact and theoretical groundedness of the paper! Your insightful feedback and valuable references really helps increase the quality of the paper and provide better position of this method. We address your concerns and questions as follows.
> ### Soundedness
> * We really appreciate your thoughtful attention on the **unconfoundedness assumption** under abstention. We would like to address the following remarks on this assumption.
>   * **Role.** We totally agree that unconfoundedness assumption worths more discussion when considering abstention in causal decision-making. As formalized in Assumption 3, it assumes that $(Y(0),Y(1))\perp T|X$, i.e., the potential outcomes are independent of the treatment $T$ given covariate $X$. Also made in previous works on treatment responder classifications, unconfoundedness assumption is a **fundamental assumption** to identify the conditional average treatment effect $\tau(x)=E[Y(1)-Y(-1)\mid X=x]$ so as to estimate the conditional responding probability $\rho(X)=P(R=1\mid X)$ through Proposition 2.
>   * **Further illustration on example.** We would like to add the follows remarks to clarify the meaning of our example .
>     * First, we would like to point out that the motivation of abstention **well exists** as individuals still have different levels of uncertainty when unconfoundedness holds. The disparity in uncertainty comes from the **non-uniform** distribution of covariates. Those sample points with sparse data reasonably have larger uncertainty (less efficiency) level comparing with the sample points with rich data.
>     * In practice, the difference of uncertainty and hence the motivation of abstention **always exists** no matter how many covariates we collect. Therefore, unconfoundedness assumption is classically regarded as an ideal but acceptable assumption to imply that enough information has been collected by the practitioner in the experiment.
>     * Meanwhile, we are well aware that the clinical decision-making example in introduction is a bit misleading in terms of the unconfoundedness concern. For better alignment to our framework, we have edited the example to consider a case when the confounders have been sufficiently collected in practice. However, doctors may observe **rare lesions** for some patients, and the uncertainty on these patients to benefit from the surgery may be significantly higher than others since there has been few samples with such lesion. Abstention learning, in this case, enables practitioners to select these patients for further actions such as resorting to external information. We have updated this example in the revised version.
>   * **Extension when unconfoundedness violates.** Meanwhile, we appreciate the reviewer for raising the concern that unconfoundedness may be violated in some situations, for example, when the rule is of high uncertainty because of unobserved confounders such as undetermined lesions from the scan. To address this concern, we further extend our methods to accommodate possible violation on unconfondedness assumption through constructing sensitivity bound on the CATE $\tau(x)$ and thereon minimizing the upper bound on loss function $L_{\theta,\delta}(f)$.
>     * Let $U$ be the observed confounder. Then in the absense of observed confounders, Assumption 3 is shifted to $(Y(0),Y(1))\perp T|X,U$. Suppose the sensitivity bound constructed on $\tau(x)$ is $\tau(x)\in[\tau^L(x),\tau^U(x)].$ Then from Proposition 2, under Assumption 1,2,4, we have $\rho(x)\in [\tau^L(x)/2,\tau^U(x)/2].$ Therefore, from Proposition 3, we have $$V_f(x)\le U_f(X):=\frac{1-f(x)}{2}\cdot(1-\theta)\tau^U(x)+\frac{1+f(x)}{2}\cdot\theta(1-\rho^L(x))$$ and hence we have
>      $L\_{\theta,\delta}(f)\le \sup_{\beta\in\mathbb{R}} \\{\beta+\frac{1}{1-\delta}E(U\_f(X)-\beta)_-\\}.$ And the treatment responder is selected through minimizing the upper bound above with the same procedure as in Algorithm 1.
>     * In practice, we adopt the plug-in estimator on the following sensitivity bound inspired by [3]:
>       $$\tau(x) \in [\pi(x)\mu_{1,1}(x)-1-(1-\pi(x))\mu_{-1,-1}(x),\pi(x)\mu_{1,1}(x)+1-(1-\pi(x))\mu_{-1,-1}(x)],$$ where $\mu_{t,t'}(x)=E(Y(t)\mid T=t',X=x).$ In our experiments, we use _Twins_mono dataset_, with other settings same as before, and compare our methods with the most competitive baselines.
>
>        |**Model**|**f-regret (In)**|**f-regret (Out)**|
>        |-|-|-|
>        |**TRECA**| **0.003 ± 0.002**|**0.003 ± 0.003**|
>        |**TRECA (w/o unconf.)**| **0.004 ± 0.002**|**0.004 ± 0.002**|
>        |CTRL|0.004 ± 0.001|0.006 ± 0.002|
>        |DragonNet|0.084 ± 0.166| 0.087 ± 0.163|
>        |DeRCFR|0.098 ± 0.191|0.103 ± 0.190|
>        |CFRNet|0.018 ± 0.030|0.016 ± 0.026|
>
> The results show that with slightly increase in regret due to relaxation in assumption, TRECA still achieves notably lower regret classifying treatment responders on retained samples in relaxation of unconfoundedness assumption.

---

> ### Author Response · Authors · 2025-11-24
>
> * **Soundedness (continued)** In the revised manuscript, we have added discussion on Assumption 3, articulated the example in the introduction, and appended a discussion on the extension of our method to possible violation on unconfoundedness assumption in section 3.3, which the supplementary experimental results provided in Section 4. Looking forward to know whether your concerns on the unconfoundedness assumption have been addressed!
>
> ### Clarity
>
> We sincerely appreciate your the careful reading and helpful comments regarding the consistency of notations. We acknowledge that the notation in the paper is dense due to the need for a rigorous theoretical framework. In the revised manuscript, we have thoroughly double-checked and aligned all notations for consistency. Additionally, we have added a summary table in Appendix C to clarify the key notations used throughout the paper. Please let us know if there is any notation that requires further clarification, thank you!
>
> ### Empirical Evaluation
> Thank you for this insightful suggestion regarding the abstention strategy! Following your recommendation, we have added experiments to comprace TRECA and baseline methods under **uncertainty-based** abstention strategy. For each model, we computed the conditional risk defined in Proposition 2 for every sample and ranked the samples based on the conditional risk accordingly. We then **abstain the top $\delta$ proportion of samples** with the highest uncertainty (conditional risk) and evaluated each model based solely on its performance on the remaining samples. All models were trained on the Twins_mono dataset with the same experimental settings (**$\theta$=0.5, abstention rate $\delta$=0.1**) and the same hardware configuration to ensure strict comparability. Furthermore, we evaluate the performance through the f-regret on retained samples **f-regret**, abstained samples **f-regret_abs**, and entire samples **f-regret_overall** respectively. The updated results are summarized in the table below:
>
> | **Model** | **f-regret** | **f-regret_abs** | **f-regret_overall** | **f-regret** | **f-regret_abs** | **f-regret_overall** |
> |:--|:--:|:--:|:--:|:--:|:--:|:--:|
> |  | *(In-sample)* | *(In-sample)* | *(In-sample)* | *(Out-sample)* | *(Out-sample)* | *(Out-sample)* |
> | **TRECA** | **0.003 ± 0.002** | 0.352 ± 0.171 | 0.039 ± 0.017 | **0.003 ± 0.003** | 0.322 ± 0.166 | 0.036 ± 0.017 |
> | CTRL | 0.004 ± 0.001 | 0.273 ± 0.029 | 0.085 ± 0.009 | 0.006 ± 0.002 | 0.283 ± 0.040 | 0.089 ± 0.012 |
> | DragonNet | 0.084 ± 0.166 | 0.370 ± 0.190 | 0.279 ± 0.207 | 0.087 ± 0.163 | 0.372 ± 0.196 | 0.276 ± 0.211 |
> | DeRCFR | 0.098 ± 0.191 | 0.470 ± 0.032 | 0.456 ± 0.114 | 0.103 ± 0.190 | 0.481 ± 0.020 | 0.460 ± 0.117 |
> | CFRNet | 0.018 ± 0.030 | 0.063 ± 0.065 | 0.023 ± 0.032 | 0.016 ± 0.026 | 0.065 ± 0.064 | 0.021 ± 0.027 |
>
> The results show that under uncertainty-based abstention strategy, **TRECA still outperforms the baseline method**. In particular, TRECA achieves **the lowest f-regret**, indicating that our method more effectively improves performance on the retained samples by appropriately abstaining on uncertain cases. These findings further validate the robustness and effectiveness of TRECA under consistent abstention strategies. We have included these new results and the corresponding analysis in the revised manuscript.

---

> ### Author Response · Authors · 2025-11-24
>
> ### Relation to Previous Works
>
> * Thank you for pointing out the related works which do provide a better position of our work. We add the following discussion on the connection between our work and recent literature on abstention and causal deferral, especially the works that you mentioned.
>   * The work [1] introduces a counterfactual evaluation metric for abstaining classifiers, focusing on evaluation rather than learning. In contrast, our work proposes a new abstention-enabled causal classifier that learns optimal rules via doubly robust estimation and connects abstention to Conditional Value at Risk (CVaR).
>   * Compared to [2,3], which study deferral systems where uncertain decisions are delegated to a **secondary agent** which is typically a human, the goals are similar in terms of minimizing risk under constraint on the proportion of deferred / abstained samples. However, a key difference is that our work focuses on treatment responder selection, a causal decision-making task while [2,3] aim at different tasks of learning the causal effect under RD design or outcome reward maximization. Since treatment responder $R=I(Y(1)>Y(0))$ involves the joint distribution of potential outcomes, it arouses extra challenges in identification and estimation. Adapting abstention under **full / partial identification** on the treatment reponsder is one of the main contributions of our paper. Moreover, our method can also adapt to causal effect estimation task, in which our method can be treated in parallel with [2] by tackling the problem in a different perspective from the RD design.
>
> * **Comparison with _Franc et al., 2023_.** As both our method and _Franc et al., 2023_ lie in the abstention learning framework, the abstention rules are both in the form of threshold functions, while our work features the following key differences in causal settings.
>   * (1) Our work explores the uncertainty quantity referred as **conditional risk** to be thresholded under treatment responders classification. It is not a straightforward extension from non-causal settings given that the conditional risk relies on counterfactual outcomes that can not be observed.
>   * (2) Our work establishes a thorough framework to identify the conditional risk under practical assumptions. It also dicusses the case when conditional risk can only be partially identifed due to possible violation on monotonicity or unconfoundedness assumption. From the abstention learning perspective, it can be viewed as an extension from existing methods to make abstention when we can obtain a bound instead an accurate estimation on the uncertainty per se.
>   * (3) Our work reveals that loss minimization under abstention can be re-formulated as the CVaR loss minimization with respect to the conditional risk, which enables us to develop doubly robust methods in abstention learning.
>
> We hope this clarification addresses your concerns and thanks again for your thoughtful and constructive feedback. Looking forward to further discussion!

---

> > ### Comment · Reviewer_YhGp · 2025-11-27
> >
> > I thank the authors for their clarifications. I will increase my score to a full accept.

---

> > > ### Author Response · Authors · 2025-11-28
> > >
> > > We are delighted to have your concerns addressed. Thanks again for your valuable feedback and recognization on our efforts!

---

### Official Review · Reviewer_n12V · 2025-10-31

**Soundness:** 3
**Presentation:** 1
**Contribution:** 3
**Rating:** 4
**Confidence:** 3

**Summary:**

This work examines the problem of treatment responder classification when abstention is allowed. In Theorem 1, the authors derive a connection between their problem and another minimization problem that involves a well-known quantity, the Conditional Value at Risk (CVaR). Subsequently, they provide convergence guarantees for their methods both with and without a monotonicity assumption. Finally, they conduct a comprehensive experimental evaluation comparing their methods with several others on two real-world datasets.

**Strengths:**

1) Despite this problem is well known in Online Learning literature (see i.e. https://arxiv.org/pdf/1905.09561, https://arxiv.org/abs/2305.10564 and references therein) it’s yet unexplored in the causal inference literature.
2) The authors derived a nice connection of causal classification with abstention with Condition Value at Risk (CVaR), a well known quantity.
3) The authors propose two algorithms to solve this problem and establish theoretical results guaranteeing their asymptotic convergence.
4) The authors compare their experimental results against numerous benchmarks, and their methods demonstrate superior performance.

**Weaknesses:**

1. I believe several sections of the text require revision.
2. The authors should add an introductory paragraph explaining the notation used throughout the paper (i.e., the min and max operators, the O_p notation, etc.).
3. In lines 132-134, the authors claim that the constraint formulation ensures generalization performance for their classifier. I believe this section requires clarification: generalization is guaranteed only for the subsets of P where the algorithm does not abstain. In other regions, performance may be arbitrarily poor.
4. In Theorem 1, it is unclear why the abstention rule r^\star necessarily belongs to the hypothesis set \mathcal{R}.
5. I suggest that after introducing Assumptions 4 and 5, the authors include a brief discussion characterizing the strength of these assumptions and providing examples that satisfy them. For instance, I question whether Assumption 5 (monotonicity) holds in medical scenarios. Specifically, prescribing a drug instead of a placebo to a non-patient does not always produce positive effects.
6. In Theorem 2, the authors should explicitly state in the main text the dependence on problem-specific quantities such as \delta and c, and clarify why this result advances the existing literature. For example, the conditions under which the \epsilon_n quantity vanishes remain unclear.
7. Algorithm 1 is clearly presented and well-written; however, no theoretical guarantee (theorem) establishes its convergence or convergence rate.
8. In lines 312-313, the authors state that it is possible to derive differentiable upper and lower bounds for \rho(x) with respect to \tau, yet they subsequently employ \max{\tau/2, 0}, a function that is non-differentiable in \tau at 0. Why does this non-differentiability at zero not pose a problem?
9. In Theorem 4, the authors require that the predictor hypothesis class \mathcal{F} have VC dimension larger than n. What about simpler classes, such as linear or logistic models? Do you have experimental or theoretical evidence demonstrating the performance of your method in these cases? I would expect the theoretical results to be parameterized by statistical quantities characterizing both the predictor class \mathcal{F} and the abstention class \mathcal{R} (such as VCdim(F,R)).
10. In the experimental section (line 392), it remains unclear how the authors used the MLP to construct both the predictor and the rejector. How many parameters does this MLP contain? Additionally, why do the benchmark methods abstain at random locations rather than at the same points where your method abstains?

**Questions:**

1) See the above section.
2) I believe a natural extension would be to consider the dual problem: minimizing the number of abstentions subject to maintaining low misclassification rates on the training set. Consider a scenario where you have a stream of patients and wish to assign the correct treatment to each one without abstaining from too many cases.

Typos: line 104 Y_i(1) ,  Y(-1)

---

> ### Author Response · Authors · 2025-11-24
>
> Thank you for your appreciation on the **contribution** and **soundedness** of this work! Meanwhile, we are well aware that it has not been a completely smooth experience to comprehend the details of this work owning to the dense notations in order to provide a rigorous statement on the framework and theories. Taking clarity enhancement as one of the prior goals in revision, we have done multiple changes to the manuscript to summarize the notations and **add key discussions** on multiple aspects on our method. Below, we respond to each concern in the same order raised.
>
> > Q1. Several sections / typos of the text require revision.
>
> Thank you for pointing this out! We have carefully checked the entire manuscript and correct identified typos. We have also added diagrams, extended discussions and reviews, made further explanation on the framework as well as provided additional experimental results to ensure clarity and consistency. We sincerely hope that these revisions could provide you a better understanding to our method and a smoother reading experience!
>
>
> > Q2. The authors should add an introductory paragraph explaining the notation used throughout the paper (i.e., the min and max operators, the O_p notation, etc.).
>
> Thanks for your idea. To facilitate clarity in notation throughout the paper, we have added a table to summarize key notations and their meanings in Appendix C. Some important notations are:
>
> | **Notation**| **Meaning**|
> |-|-|
> | $\min,\max$| Minimum / Maximum operator.|
> | $O_p(\cdot)$| Probabilistic order.|
> | $o_p(\cdot)$| Converge in probability.|
> | $\mathcal{F}$| Hypothesis class for the predictor $f$.|
> | $\mathcal{R}$| Hypothesis class for the rejector $r$.  |
> | $f(x)$| Predictor determining whether an individual is a treatment responder.|
> | $r(x)$ | Rejector (abstention rule); $r(x)=1$ indicates abstention.|
> |$T$| Treatment assignment. |
> | $Y$| Observed outcome.|
> | $R$| Responder indicator.|
> | $(u)^-=\min(u,0)$ | Negative part operator. |
> | $\delta$| Abstention rate. |
> | $\theta$  | Weighting parameter on misclassification types. |
>
> > Q3. In lines 132-134, the authors claim that the constraint formulation ensures generalization performance for their classifier. I believe this section requires clarification: generalization is guaranteed only for the subsets of P where the algorithm does not abstain. In other regions, performance may be arbitrarily poor.
>
> Thanks for your insightful feedback, and this is exactly what we mean. A big advantage of our method is that it enables accurate responder classification on **retained sample** at an affordable cost of the learning accuracy on abstained samples.
> * To further illustrate such key insight, we have added experiments to evaluate the model performance on retained samples and abstained samples as follows. We tracked the f-regrets defined in evaluation metrics which measured the classification accuracy on retained samples **f-regret**, abstained samples **f-regret_abs**, and on all samples **f-regret_overall** respectively. For comparison, we also evaluated one of the most competitive baselines CTRL, another treatment responder selection method  using thresholding abstention strategy as introduced in Appendix E of the revised manuscript. The abstention rate $\delta=0.1$, $\theta=0.5$ and we follow the same configuration settings as those in the main experiment. The result is shown as follows.
>
> |Model|f-regret|f-regret_abs|f-regret_overall|**f-regret/f-regret_abs**|
> |-|-|-|-|-|
> |**TRECA(ours)**|**0.0017**|0.3333|0.0357|0.0051|
> |CTRL|0.0035|0.1970|0.0233|0.0178|
>
> Results show that **TRECA** achieves **51.4\%** decrease in **f-regret** on retained samples comparing with CTRL as well as a comparable performance on the f-regret on entire samples. The **f-regret** on retained samples is **33.3\%** comparing to that on entire samples, indicating **TRECA prioritizes optimizing accuracy on the retained samples with at an affordable cost on abstained samples**.

---

> ### Author Response · Authors · 2025-11-24
>
> > Q4. In Theorem 1, it is unclear why the abstention rule r^\star necessarily belongs to the hypothesis set \mathcal{R}.
>
> * This is a good question. As illustrated in Figure 1 in the revised manuscript, in our framework, the optimal predictor and rejector can be formulated in a sequential order. The optimal predictor $f^\*$ can be formulated as the minimizer of CVaR loss $f^\*=\arg\min_{f\in\mathcal{F}} L_{\theta,\delta}(f),$ and the optimal abstention rule is determined through thresholding the conditional risk based on the trained predictor $r^\*(x)=I(V_{f^\*}(x)> F^{-1}\_{V_{f^\*}}(1-\delta)).$ The optimality is shown in Theorem 1. This inpires a sequential training process, which in the first stage we learn the predictor $\hat{f}=\arg\min_{f\in\mathcal{F}} \hat{L}\_{\theta,\delta}(f),$ and then determine the abstention rule through thresholding $\hat{r}(x)=I(V_{\hat{f}}(X)> F^{-1}_{V\_{\hat{f}}}(1-\delta)).$
> * The condition $r^\*\in\mathcal{R}$ ensures that the optimal abstention rule in thresholding form is contained in the hypothesis set. If $r^\*\notin \mathcal{R}$, one might need extra control on the systematic error between the best abstention strategy $\hat{r}$ in hypothesis set $\mathcal{R}$ and optimal thresholding rule $I(V_{f^\*}(X)> F^{-1}\_{V_{f^\*}}(1-\delta))$.
> * To include $r^\*$ in $\mathcal{R}$, it is sufficient that the class of all thresholding functions $\\{ g(X)=I(V\_f(x)>t)\mid f\in\mathcal{F},t\in \mathbb{R} \\}$ is contained in $\mathcal{R}$, which has the same level of complexity with $\mathcal{F}$ and the condition is satisfied in practice. We have added the discussion in the revised manuscript. Thanks for your valuable remark!
>
> > Q5. I suggest that after introducing Assumptions 4 and 5, the authors include a brief discussion characterizing the strength of these assumptions and providing examples that satisfy them. For instance, I question whether Assumption 5 (monotonicity) holds in medical scenarios. Specifically, prescribing a drug instead of a placebo to a non-patient does not always produce positive effects.
>
> Thanks for your concern. We have extended the following explanations and examples on Assumptions 4 (continuity) and 5 (monotonicity).
> - Assumption 4 (continuity) assumes that the conditional risk $V_f(X)$ has bounded probability density function (pdf). The existence of pdf ensures that the quantile $F_{V_f}^{-1}(1-\delta)$ leveraged to quantify the abstained samples in Eq. (4) is unique for $\delta\in(0,1)$, backing up the uniqueness of abstention rule. The continuity on pdf, also made in Lemma EC.2 in Kallus (2023), bridges the threshold difference and conditional risk gap as in the proof on Theorem 3 after Eq. (12) in appendix. The boundedness condition ensures that Eq. (14) holds
> $$
> \mathbb{P}(|V\_f(X) - \beta^\*| \leq |\hat{\beta}_U - \beta^\*|)\leq 2c|\hat{\beta}_U - \beta^\*|
> $$
> so as to bound the residuals in derivation of Theorem 1. Assumption 4 is satisfied for general covariates, e.g., when $X=(X_1,X_2)$, where $X_1$ are discrete random variables with finite classes, $X_2$ are continuous variables with bounded density. The conditional risk $V_f(X)$ is a differentiable function in terms of $X_2$.
>
> - In drug-treatment example, Assumption 5 (Monotonicity) ensures that the drug will not have a negative effect on the drug-takers. It is indeed an asssumption that requires scrutinity in application. As discussed in the literature review, monotonicity is a classical assumption to identify the treatment responder. We also propose an extension of our method to learn abstention rule **undr violation of monotonicity assumption** in Section 3.2. When the monotonicity does not hold, we can **partially identify** the responding probability by deriving bound $\rho(x)\in(\rho^L(x),\rho^U(x)).$ This enables us to construct tight upper bound on conditional risk $V_f(x)$ and hence the loss function from Proposition 3. The predictor is learned through minimizing a debiased estimator on the upper bound of loss function derived in Eq. (7).
>
> In the revised manuscript, we have re-organzied the discussion **added further explantion** on Assumption 4 and highlighted that our methods can accommodate violation on Assumption 5.

---

> ### Author Response · Authors · 2025-11-24
>
> > Q6. In Theorem 2, the authors should explicitly state in the main text the dependence on problem-specific quantities such as $\delta$ and $c$, and clarify why this result advances the existing literature. For example, the conditions under which the $\epsilon_n$ quantity vanishes remain unclear.
>
> We greatly appreciate your careful feedback on the details of the theoretical part, and we would like to add the following discussion to address your concern.
>
> * **Dependence on quantity.** We thank the reviewer by raising concern on the dependence of the convergence rate to the problem-specific quantities in Theorem 2, i.e., $c$, $\underline{\pi}$ and $\delta$.
>   * From Eq.(13) and (14) in the proof of Theorem 2, the convergence rate is linear to $c$.
>   * The parameters $\underline{\pi},\delta$ are applied to bound Eq. (11), and the bound is linear to $\overline{\pi}^{-1}$ and $(1-\delta)^{-1}$ from the proof of Theorem 2.
>   * Therefore, the convergence will not be sensitive the extreme choices of the quantities. We have added the discussion on parameter dependence after Theorem 2.
> * **Advance on existing literature.** The robust theory developed in our work is an extension from existing literatures on double robustness theory on causal effect estimators by taking abstenion and decision-making into account. This involves discontinuous elements such as indicator $I(V_{f}(X;\tau)\le \beta)$ in the loss function, which requires extra techniques to prove the convergence rate. For example, in the proof of Theorem 2 we apply H\\"{o}lder's inequality
> $$
> \mathbb{E}[|V_f(X;\tau)-V_f(X;\hat{\tau})|I[|V_f(X;\tau)-\hat{\beta}|\leq t]]\le \|V_f(X;\tau)-V_f(X;\hat{\tau})\|_q\mathbb{P}(|V_f(X;\tau)-\hat{\beta}|\leq t)^{(q - 1)/q}
> $$ to separate the gaps between conditional risks and the density function.
> * **Vanish on $\epsilon_n$.** The error on nuisance parameter $\epsilon_n$ vanishes with consistent estimators $\hat{\mu}$ and $\hat{\pi}$, which is satisfied for MLP-based estimators applied in our framework [1], as well as other well-known regression-based propensity score estimators such as kernel regression [2].
>
> We have modified the formulation of Theorem 2 as well as added further discussion in the revised text. Thanks again for the suggestion!
>
> [1] A. Farago et al., Strong universal consistency of neural network classifiers. IEEE Transactions on Information Theory, 1993.
>
> [2] Hirano K, Imbens G W, Ridder G. Efficient estimation of average treatment effects using the estimated propensity score. Econometrica, 2003.
>
> > Q7. Discussion on the convergence on Algorithm 1.
>
> Thanks for your concern. Our algorithm essentially applies gradient descent, a well-known optimization method with convergence rate at $O(T^{-1/2})$ with $T$ iterations. On the experimental side, the algorithm converges empirically in all cases in our experiments. The following table compares the convergence time between our method **TRECA** and other baseline methods. Results show that our method has medium level of speed in convergence, which provides a favorable balance between efficiency and accuracy.
>
> |Model|Time(seconds)|Rate|
> |-|-|-|
> |DragonNet|4.30s|10.19%|
> |DESCN|5.18s|12.28%|
> |ESCFR|8.97s|21.27%|
> |CFRISW|12.64s|29.97%|
> |**TRECA(ours)**|42.18s|100%|
> |CFRNet|86.83s|205.86%|
> |DeRCFR|100.89s|239.19%|
> |CTRL|134.20s|318.16%|
> |CEVAE|183.75s|435.63%|
>
> > Q8. In lines 312-313, the authors state that it is possible to derive differentiable upper and lower bounds for $\rho(x)$ with respect to $\tau$, yet they subsequently employ $\max\\{\tau/2, 0\\}$, a function that is non-differentiable in $\tau$ at 0. Why does this non-differentiability at zero not pose a problem?
>
> Thank you for noting this. In experiments, the ReLU function $\max\\{x, 0\\}$ is substituted by the well-known differentiable approximation Softplus function $log(1+exp(x))$ in surrogate, which is also applied, for example, in learning individual treatment rules such as [1]. We have supplemented the details in the appendix.
>
> [1] Zhao et al., Estimating Individualized Treatment Rules Using Outcome Weighted Learning, JASA 2012.

---

> ### Author Response · Authors · 2025-11-24
>
> > Q9. In Theorem 4, the authors require that the predictor hypothesis class $\mathcal{F}$ have VC dimension larger than n. What about simpler classes, such as linear or logistic models? Do you have experimental or theoretical evidence demonstrating the performance of your method in these cases? [Q9.1] I would expect the theoretical results to be parameterized by statistical quantities characterizing both the predictor class $\mathcal{F}$ and the abstention class $\mathcal{R}$ (such as VCdim(F,R)). [Q9.2]
>
> * [A9.1] Thank you for pointing this out. To examine whether impact of the space complexity in hypothesis class of predictors $\mathcal{F}$ on model performance, we add an experiment to evaluate the performance of our method under different layers of predictor networks that correspond to different levels of space complexity. We vary the number of layers in the predictor network and evaluate architectures with 4, 10, 30, and 50 layers under the settings of $\theta=0.5$ and an abstention rate of $\delta=0.1$ on the _Twins\_mono_ dataset. We also compare the performance with the predictor trained from logistic regression, i.e., $P(f(x)=1)=logit(x^\top\alpha).$ The results are shown below.
>
>     |layer of f|4|10| 30| 50|logistic|
>     |-|-|-|-|-|-|
>     | f-regret| 0.0035 | 0.0035   | 0.0034| 0.0032| 0.0547|
>     | f-regret_abs| 0.3485 | 0.2273| 0.0758   | 0.0530 | 0.2930|
>     | f-regret_overall| 0.0388 | 0.0264 | 0.0109| 0.0031 |0.1308|
>
>  * The results indicate that increasing the depth of the predictor network improves model performance. Specifically, the regret on the retained samples (**f-regret**) exhibits a **slight but steady decrease** as the depth grows from 4 to 50 layers, together with improvements on the regret on abstained samples (**f-regret_abs**) and the overall sample (**f-regret_overall**). The **f-regret** using a neural network is smaller than that using logistic regression. These findings suggest that deeper predictor networks can better capture the uncertainty structure and further enhance the effectiveness of the abstention mechanism.
>
>  * In the proof of Theorem 4, the condition $VC(\mathcal{F})\ge n$ is used to guarantee that the optimal learnable predictor $\hat{f}_M$ classifies the responder correctly for **deterministic samples under partial identification**, i.e., $\hat{f}_M(x)=-1$ for $\rho^U(x;\hat{\tau})\le\theta$ and $\hat{f}_M(x)=1$ for $\rho^L(x;\hat{\tau})>\theta.$ This enables us to bound the gap on rejector $P(\hat{r}_M(X)\neq r^\*(X))$ through the predictors $P(\hat{f}_M(X)\neq f^\*(X))$. We have added further discussion on the conditions and substitute the condition on VC dimension with the **weaker condition** above while place the VC condition as a special case in the revised manuscript.
>
> * [A9.2] We appreciate your insightful question. The space complexity bound in Theorem 2 is specifically tailored to our learning framework. As explained in **Q4**, the loss minimization with a constraint on the abstention rate is reformulated as minimizing the CVaR-based loss $L_{\theta,\delta}(f)$ with respect to $f$, and the abstention rule $r$ is subsequently determined by thresholding the conditional risk of the learned predictor $r^\*(x)=I(V_{f^\*}(X)> F^{-1}\_{V_{f^\*}}(1-\delta)).$ Consequently, the complexity analysis depends only on the hypothesis class $\mathcal{F}$, since the abstention rule $r$ is directly estimated from thresholding the conditinal risk given the trained predictor.
>
> > Q10. In the experimental section (line 392), it remains unclear how the authors used the MLP to construct both the predictor and the rejector. How many parameters does this MLP contain? [Q10.1] Additionally, why do the benchmark methods abstain at random locations rather than at the same points where your method abstains? [Q10.2]
>
> * [A10.1] Thanks for your concern. In the TRECA model, the representation network consists of two hidden layers, the propensity network contains three hidden layers, and the outcome network includes two hidden layers. All hidden layers are set to a dimensionality of 16 with ELU activation function. To clearly present our configuration, we also provide a table summarizing the configuration settings for the TRECA model.
>
>     |Parameter|Value|
>     |-|-|
>     |CPU|Intel Core i5-1155G7|
>     |batch size|500|
>     |epoch|1000|
>     |learning rate|0.0005|
>     |Representation network hidden layer|2|
>     |propensity network hiddenlayer|3|
>     |outcome network hidden layer|2|
>     |representation network hidden layer|2|
>     |hidden dimension|16|
>     |activation for hidden layer|ELU|
>     |activation for output layer|Sigmoid|
>     |optimizer|Adam|

---

> ### Author Response · Authors · 2025-11-24
>
> * [A10.2] Great appreciation for your insightful suggestion on comparison with baseline methods using non-random abstention strategy!
>   * Since one of the main advantages of our method is to select samples to be abstained to enhance performance on retained samples, the abstention strategy itself is an important contributioon of our method, while other methods on treatment reposnder classification or causal effect estimation do not provide us with a clear abstention strategy.
>
>   * Following the reviewer's suggestion on a more advanced abstention strategy among benchmark methods for better comparison, we also applied a thresholding abstention strategy on baseline methods. For baseline estimators on CATE $\hat{\tau}(x)$, following the derivation in Proposition 2, we compute the conditional risk $$\hat{V}_f(x)=\frac{1}{2}f(x)\{\theta-\hat{\rho}(x)\}+\frac{1}{2}\{\theta+(1-2\theta)\hat{\rho}(x)\}$$ with $\hat{\rho}(x)=\hat{\tau}(x)/2$, and abstain the samples with $100\delta\%$ highest conditional risk. Under this same setting as in the main experiments, the results inshown as follows. **f-regret** means the regret on retained samples, while **f-regret_abs** and **f-regret_overall** indicate the regrets on abstained samples and all samples respectively.
>
>       | **Model** | **f-regret** | **f-regret_abs** | **f-regret_overall** | **f-regret** | **f-regret_abs** | **f-regret_overall** |
>       |:--|:--:|:--:|:--:|:--:|:--:|:--:|
>       |  | *(In-sample)* | *(In-sample)* | *(In-sample)* | *(Out-sample)* | *(Out-sample)* | *(Out-sample)* |
>       | **TRECA** | **0.003 ± 0.002** | 0.352 ± 0.171 | 0.039 ± 0.017 | **0.003 ± 0.003** | 0.322 ± 0.166 | 0.036 ± 0.017 |
>       | CTRL | 0.004 ± 0.001 | 0.273 ± 0.029 | 0.085 ± 0.009 | 0.006 ± 0.002 | 0.283 ± 0.040 | 0.089 ± 0.012 |
>       | DragonNet | 0.084 ± 0.166 | 0.370 ± 0.190 | 0.279 ± 0.207 | 0.087 ± 0.163 | 0.372 ± 0.196 | 0.276 ± 0.211 |
>       | DeRCFR | 0.098 ± 0.191 | 0.470 ± 0.032 | 0.456 ± 0.114 | 0.103 ± 0.190 | 0.481 ± 0.020 | 0.460 ± 0.117 |
>       | CFRNet | 0.018 ± 0.030 | 0.063 ± 0.065 | 0.023 ± 0.032 | 0.016 ± 0.026 | 0.065 ± 0.064 | 0.021 ± 0.027 |
>
> It can be observed that **TRECA achieves the lowest regret on retained samples**, further demonstrating that TRECA improves the performance on the retained samples without much compliance on the regret on the entire samples.
>
> > Q.11 I believe a natural extension would be to consider the dual problem: minimizing the number of abstentions subject to maintaining low misclassification rates on the training set. Consider a scenario where you have a stream of patients and wish to assign the correct treatment to each one without abstaining from too many cases.
>
> This is indeed an interesting extension! The dual problem—minimizing abstention subject to bounded misclassification—also corresponds to a constrained optimization but with misclassification guarantees. Our current formulation minimizes misclassification risk under an abstention constraint. The dual formulation,
> $$
> \min_{r} E[r(X)] \quad \text{s.t.} \quad \tilde{L}\_{\theta}(f,r) \le \varepsilon,
> $$
> represents minimizing abstention subject to an upper bound on error where excessive abstention is undesirable. From the proof on Theorem 1, $$\tilde{L}\_{\theta}(f,r)=\int_{x\in\mathcal{X}} V\_f(x)p(x\mid r(x)=0)P(r(x)=0)\\,dx.$$ Therefore,
> given $f^\*$, the optimal abstention rule $r^\*$ that minimizes $E[r(X)]$ is the rule that abstain samples with the highest uncertainty until $\tilde{L}\_{\theta}(f,r) \le \varepsilon$ for retained samples. This yields a thresholding rule $r^\*(x)=I(V\_f(x)>\beta'),$ with $$\beta'=\sup_\beta \int_{x\in\mathcal{X}} V\_f(x)p(x\mid V\_f(x)\le\beta)P(V\_f(x)\le\beta)\\,dx.$$ Therefore, our method can adapt to this case by choosing a different threshold such that the loss constraint on retained samples is satisfied. Thank you for proposing this meaningful and applicable dual problem. We have added the discussion in the revised manuscript.
>
> Please let us know if we have properly addressed your questions and we are more than happy to discuss more!

---

### Official Review · Reviewer_EWwE · 2025-11-01

**Soundness:** 2
**Presentation:** 2
**Contribution:** 2
**Rating:** 4
**Confidence:** 3

**Summary:**

This paper investigates the treatment responder classification problem, aiming to develop a classification rule that provides predictions for most individuals while permitting a small subset of samples to abstain from decision-making, thereby minimizing overall misclassification risk. The authors first establish the connection between causal misclassification risk and Conditional Value at Risk (CVaR). Building on this insight, they propose a doubly robust classification method named TRECA, which jointly estimates the propensity score and outcome models to achieve stable performance even under weak convergence conditions. Furthermore, they introduce an enhanced variant, TRECA+, which relaxes the monotonicity assumption to improve adaptability in non-monotonic scenarios. Comprehensive experiments on two real-world datasets (Twins and Jobs), along with their monotonicity-transformed counterparts, demonstrate that both TRECA and TRECA+ surpass various baseline methods in classification accuracy and stability.

**Strengths:**

(1) The paper formalizes the modeling of the Treatment responder classification problem based on the causal inference framework, with a systematic derivation process that possesses theoretical depth.

(2) The paper introduces an abstention decision mechanism into the causal responder classification task, enhancing the model's practicality from the perspectives of risk minimization and decision theory, with promising application prospects.

**Weaknesses:**

(1) The paper includes only Figure 1; however, Figure 3 is referenced on line 430 of page 8 in the main text, and Figure 5 is mentioned in the appendix. It is recommended that the authors carefully check and supplement the missing content.

(2) In the introduction section, the paper provides a clear overview of the treatment responder classification and abstention selection problems, but it does not adequately elaborate on the limitations of existing methods, making it difficult to distinguish the differences between the proposed method and other approaches.

(3) The methods section primarily focuses on mathematical derivations (such as loss functions, theorems, assumptions, and proofs), lacking an intuitive description of the overall model structure. For example, the paper mentions the predictor and rejector, but does not clearly explain their training process.

(4) The paper lacks an overall framework diagram to intuitively illustrate the structure and process of the method.

(5) The methods section of this paper lacks discussion on time complexity or space complexity. It is recommended that the authors include corresponding analyses, particularly on the computational complexity in key steps such as the joint estimation of the propensity score model and outcome model, as well as the doubly robust method.

(6) The experimental setup section of the paper does not provide detailed descriptions of hardware and software configurations, such as the computing environment, GPU/CPU specifications, and so on, as well as exact parameter values, such as learning rate, batch size, optimizer type, etc.

**Questions:**

(1) Could the authors confirm whether Figures 3 and 5 are missing? Figure 3 is referenced on line 430 of page 8, and Figure 5 is mentioned in the appendix, yet only Figure 1 is provided. Could the authors either supply the missing figures or offer an explanation to enhance the completeness of the paper?

(2) The introduction clearly outlines the problems at hand, but it lacks detailed discussion on the limitations of existing methods. Could the authors elaborate on these limitations to provide a more comprehensive background?

(3) The methods section presents substantial mathematical content but does not provide an intuitive overview of the model, such as the training process for the predictor and rejector. Could the authors include such an overview to improve the clarity and accessibility of the section?

(4) There is no discussion regarding the time or space complexity of the proposed methods, particularly with respect to the joint estimation within the doubly robust framework. Could the authors address this aspect to inform readers of the computational efficiency of the approach?

(5) The experimental setup section omits key hardware and software details (e.g., computing environment, GPU/CPU specifications) as well as hyperparameters (e.g., learning rate, batch size, optimizer). Could the authors provide these details to ensure the experiments are reproducible?

---

> ### Author Response · Authors · 2025-11-24
>
> Thank you for the careful reading and inspiring feedback! We have addressed each of the raised weaknesses and questions in detail below.
>
> > [W1, Q1] The paper includes only Figure 1; however, Figure 3 is referenced on line 430 of page 8 in the main text, and Figure 5 is mentioned in the appendix. It is recommended that the authors carefully check and supplement the missing content.
>
> Thanks for your careful reading and kind remind of the incorrect referrence! The Figure 3 and Figure 5 mentioned in original version correctly point to Table 3 and Table 6 in the revised manuscript respectively. We have gone through the whole text and carefully checked the references in the revised version to ensure consistency and clarity in the revised manuscript.
>
> > [W2, Q2] In the introduction section, the paper provides a clear overview of the treatment responder classification and abstention selection problems, but it does not adequately elaborate on the limitations of existing methods, making it difficult to distinguish the differences between the proposed method and other approaches.
>
> Thanks for pointing out the lack of discussion on the limitations of existing methods in introduction. We have carefully re-organized the introduction to highlight the **limitations** of existing methods as well as the **improvements** of our methods, which are summarized as follows:
>
> * **No Abstention in Prior Treatment Rule Methods.**
>   * Most existing frameworks for individual treatment rule learning (including treatment responder classification) require a **deterministic decision for every individual**, with no provision to abstain. In other words, current individualized treatment rule methods must classify everyone as responder or non-responder, which can lead to **significant error on some cases with high uncertainty**.
>   * Our work addresses this by introducing an abstention option, allowing the model to defer decision on a small fraction of individuals for whom it is least confident, thereby improving precision on the remaining cases.
> * **Classical Abstention Learnings Faces Limitations in Causal Settings.**
>   * Traditional classification with reject option (e.g., Chow’s rule and subsequent confidence-based rejection methods) typically abstain based on a **predefined cost of rejection**, which may be hard to interpret in practice. To address these challenges, we propose a constraint-based framework to learn the abstention rule, which enhances the interpretability.
>   * Identification of misclassification loss in causal setting involves a series of nuisance parameters, which may affect the accuracy of learned treatment rule. To address this conern, we develop robust learning methods to deal with possible mis-specification on nuisance parameters.
> * **Lack of Theoretical Guarantees.**
>   * Prior robust methods in individual treatment rule learning or causal inference did not consider abstention as an option, while existing abstention learning methods also lack theoretical analysis in a causal context.
>   * Our method, by contrast, comes with **theoretical guarantees** by leveraging the doubly robust estimation property (Theorem 2) and reveals the relation between abstention to CVaR (Proposition 1), which gives us a clear criterion for optimal rejection. To the best of our knowledge, this is the first work to provide theoretically-gounded framework on treatment responder classification with abstention.
>
> Besides, we have extended the literature review and offered a more comprehensive comparison between our methods and **more than 20** relavent works covering cuasal effect estimation, indivudal treatment rule learning, abstention learning and deferral system, please kindly see the Appendix for details. We sincerely hope that these can enhance readers' understanding by providing a more comprehensive background that highlights how our proposed method differs from and improves upon previous methods.

---

> ### Author Response · Authors · 2025-11-24
>
> > [W3, Q3] The methods section primarily focuses on mathematical derivations (such as loss functions, theorems, assumptions, and proofs), lacking an intuitive description of the overall model structure. For example, the paper mentions the predictor and rejector, but does not clearly explain their training process.
>
> Great thanks for your patience on reading the paper carefully! And we apologize for possible ambiguity on the framework. To fully address your concern on a clearer explanation on the overall framework, we have **added a diagram** to demonstrate the overall training process in Figure 1, and added a **clearer summary** on the framework in Section 2 of the main text.
>
> * In summary, our method consists of the following key components:
>
>   - The **predictor** $f(x)$ outputs a classification of treatment response.
>   - The **rejector** $r(x)$ identifies abstained samples.
>   - **Nuisance parameters** such as CATE $\tau$ and propensity score $\pi$.
>
> * The overall workflow to train the predictor and rejector is summarized as follows.
>   * We first estimate nuisance functions $\hat{\mu}\_1(x)$, $\hat{\mu}\_{-1}(x)$, $\hat{\pi}(x)$ through minimizing the mean square error (MSE). This stage optimizes the standard CFR loss and yields a stable CATE estimator.
>   * Then, we learn the predictor $f$ and rejector $r$ in a sequential order.  In the first step, we train the predictor $f$ through minimizing the CVaR loss $L_{\theta,\delta}(f)=\sup_{\beta\in\mathbb{R}} \{\beta+\frac{1}{1-\delta}E(V\_f(X)-\beta)_-\}$ and estimate the conditional risk $V_f(x)=\frac{1}{2}f(x)\{\theta-\rho(x)\}+\frac{1}{2}\{\theta+(1-2\theta)\rho(x)\}$ for each individual following Proposition 2. This stage enables the model to selectively abstain on high-uncertainty samples to reduce regret on retained samples.
>   * Finally, we determine the abstention rule based on ranking the estimated conditional risk $r^\*(x)=I(V_{f^\*}(X)> F^{-1}_{V\_{f^\*}}(1-\delta)).$ The details on the algorithm are provided in Algorithm 1.
>
> > W4. The paper lacks an overall framework diagram to intuitively illustrate the structure and process of the method.
>
> Thanks for your good idea on illustrating the overall pipeline of our method through a diagram! As mentioned in the previous answer, we have added it as Figure 1 in the revised manuscript together with extra discussion on the overall framework and training process in Section 2.1.
>
> > [W5, Q4] The methods section of this paper lacks discussion on time complexity or space complexity. It is recommended that the authors include corresponding analyses, particularly on the computational complexity in key steps such as the joint estimation of the propensity score model and outcome model, as well as the doubly robust method.
>
> Thanks for your question! We have added **theoretical** as well as **experimental** analysis on both **time and space** complexity as follows.
>
> * **Time complexity.**
>   * **Theory.** Let $n$ be the sample size, $m$ batch size and let $l_\pi,l_\mu,l_\tau,l_f$ and $d_\pi,d_\mu,d_\tau,d_f$ be the number of layes and hidden dimension for the propensity, outcome, CATE and predictor networks respectively. Let $t_\phi=l_\phi d_\phi^2$ be the time complexity on a one-time forward in network $\phi.$ The analysis on time complexity of our method can be divided into three stages.
>     * Stage 1: Train the propensity net $\hat{\pi}(x)$ and outcome net $\hat{\mu}\_t(x)$ through minimizing MSE among the entire samples with size $n$, which owns the complexity of $O(n(t_\pi+2t_\mu)).$
>     * Stage 2: Train CATE $\tau$ and predictor $f$ alternatively through minimizing CVaR loss $\hat{L}\_{\theta,\delta}(f;\hat{\tau},\hat{\mu},\hat{\pi})$ in Eq. (5). Estimating CVaR loss requires calling the propensity and outcome networks, which is done $r_\beta$ rounds for each of the $n/m$ epochs, owning time complexity of $O(nm^{-1}r_\beta(t_\pi+t_\mu)).$ Then, the CATE and predictor are trained at a complexity of $O(n(t_\tau+t_f)).$ Balancing the representations on CATE estimation involves time complexity at $O(n/m\cdot m^2)=O(nm).$
>     * Stage 3: The conditional risks $V_{\hat{f}}(x)$ are computed for each individual using trained predictor. The time complexity is $O(n(t_\tau+t_f)).$
>     * In summary, the overall time complexity is $O(n[2(t_\tau+t_f)+(1+m^{-1}r_\beta)(d_\pi+d_\mu)+m])$, which is linear to all quantities, implying that the method will generally not suffer from heavy computation burden.

---

> ### Author Response · Authors · 2025-11-24
>
> * **Extended discussion on time complexity.**
>     * **Experiment.** Empirically we added experiments to measure the running time of each method under identical conditions. All models were trained on the _Twins_mono_ dataset with the same experimental settings ($\theta=0.5$, abstention rate $\delta=0.3$) and the same hardware configuration to ensure strict comparability. We recorded the total time required for each method to complete 1000 training epochs. The results are summarized in the table below:
>
>     |Model|Time(seconds)|Rate|
>     |-|-|-|
>     |DragonNet|4.30s|10.19%|
>     |DESCN|5.18s|12.28%|
>     |ESCFR|8.97s|21.27%|
>     |CFRISW|12.64s|29.97%|
>     |**TRECA(ours)**|42.18s|100%|
>     |CFRNet|86.83s|205.86%|
>     |DeRCFR|100.89s|239.19%|
>     |CTRL|134.20s|318.16%|
>     |CEVAE|183.75s|435.63%|
>
>     As shown in the table, our proposed model TRECA has **medium level of computation time**. Moreover, the models that train faster than TRECA typically exhibit larger f-regret in experiments, indicating that their efficiency comes at the cost of reduced predictive accuracy. In conclusion, TRECA maintains moderate computational complexity while achieving strong classification performance. This demonstrates that TRECA provides a favorable **balance between efficiency and accuracy**.
>
> * **Space complexity.**
>    **Theory.** Following the notations and stage division in time complexity, further define $s_\phi=d_\phi l_\phi$ to be the capacity of some network $\phi$.
>     * In Stage 1, the space complexity is equal to the network capacity times batch size $O(m(s_\pi+s_\mu)).$
>     * In Stage 2, since the model is trained through alternative gradient descent on $\tau$ and $f$ while requires calling the propensity for each individual, the space complexity for weight storage, gradient memory and propensity storage is $O(m(s_\tau+s_f)).$ Computing the representation distance in learning CATE involves $O(m^2)$ space.
>     * Learning the abstention rule in Stage 3 requires $O(1)$ extra space. In summary, the space complexity is $O(m(s_\pi+s_\mu+s_\tau+s_f)+m^2),$ which is at most quadratic to relevant parameters.
>    **Experiment.** We record and compare the system memory training our methods and baseline methods. For standardization, we set the memory for our method as 1. The following table shows that our method has a **medium level** of memory occupation. In particular, the required memory is less than that of CTRL, another treatment responder classification method using joint learning.
>
>     |Model|Memory(standardized)|
>     |-|-|
>     |CFRNet|0.5|
>     |DragonNet|0.7|
>     |CFRISW|0.7|
>     |**TRECA(ours)**|1|
>     |ESCFR|1.3|
>     |CTRL|1.4|
>     |DESCN|1.8|
>     |DeRCFR|1.8|
>     |CEVAE|2.5|
>
> > [W6, Q5] The experimental setup section of the paper does not provide detailed descriptions of hardware and software configurations, such as the computing environment, GPU/CPU specifications, and so on, as well as exact parameter values, such as learning rate, batch size, optimizer type, etc.
>
> Thanks for your kind remind on the missingness of some configuration details! We have now **double checked and added details** on the configurations to the revised manuscript. All experiments were conducted on a Core i5-1155G7 Laptop CPU. The batch size is set as 500, and training epoch set as 1000 for all models. The optimal learining rate is set as 0.0005 for our TRECA method. We employ the Adaptive Moment Estimation (Adam) optimizer during model training. In the TRECA model, the representation network consists of two hidden layers, the propensity network contains three hidden layers, and the outcome network includes two hidden layers. All hidden layers are set to a dimensionality of 16 with ELU activation function. To clearly present our configuration, we also provide a table summarizing the settings for the TRECA model.
>
> |Parameter|Value|
> |-|-|
> |CPU|Intel Core i5-1155G7|
> |batch size|500|
> |epoch|1000|
> |learning rate|0.0005|
> |Representation network hidden layer|2|
> |propensity network hiddenlayer|3|
> |outcome network hidden layer|2|
> |representation network hidden layer|2|
> |hidden dimension|16|
> |activation for hidden layer|ELU|
> |activation for output layer|Sigmoid|
> |optimizer|Adam|
>
> Thanks again for raising these valuable concerns, and we sincerely hope that the discussions and details could offer you a better experience in understanding our framework and method from macro to micro. All of these details have now been incorporated into the revised paper. Please let us know if you’d like further illustration, thank you!

---

### Author Response · Authors · 2025-11-24
**Summary on Revisions in Updated Manuscript**

We are grateful to all reviewers for their efforts and helpful comments regarding our paper. We are encouraged that reviewer `EWwE` identifies the application prospect of this work, and all three reviewers `EWwE`, `n12V` and `YhGp` find the problem interesting, the method theoretically grounded and the work has potential to be impactful.

Meanwhile, we are deeply appreciated and well aware of the concerns raised by the reviewers in the comments, which greatly help us improve the quality of our paper. In response to the key problems and suggestions, we have made the following main revisions to the manuscript, with all changes highlighted in $\textcolor{blue}{blue}$ text:

* Revise the example in the introduction for better alignment with the assumptions of our framework.
* Re-organize the expansion of the framework in Section 2, with further explanations on various aspects added.
* Add a diagram illustrating the overall framework in Figure 1.
* Re-formulate Theorem 2 and provided additional discussion in Section 3.1.
* Add further discussion on violations of the unconfoundedness assumption in Section 3.3, with experimental details in Appendix E.7.
* Add time and space complexity analysis in Section 3.4.
* Extend the literature review to include a broader comparison with over 20 related works on treatment effect estimation, individual treatment rules, abstention learning, and causal deferral systems in Appendix B.
* Summarize key notations in a table provided in Appendix C.
* Add supplementary details on the configuration and training process in Appendix E.2.
* Add additional experiments on hypothesis set complexity in Appendix E.4.
* Add experiments on time and space complexity in Appendix E.5.
* Add experiments evaluating baseline methods under uncertainty-based abstention rule in Appendix E.6.

Please review our detailed responses to each point raised. We hope that our revisions and clarifications satisfactorily address the concerns raised. We thank you again for your valuable time and expertise.

Many thanks,

Authors of #7463

---

### Author Response · Authors · 2025-12-03
**Summary on Revisions and Responses to Reviewers**

Dear AC, SAC, PC and all reviewers,

We sincerely appreciate your great efforts in evaluating our paper despite your busy schedules. We are encouraged that the only responded Reviewer `YhGp` have confirmed that our rebuttal successfully clarified the concerns with **score raised to 8** on Nov 27, while all reviewers recognizing our paper as
- the problem "very interesting", "method is theoretically grounded", "has potential to be impactful" (Reviewer `YhGp`),
- having "a systematic derivation process that possesses theoretical depth", "clearly outlines the problems at hand" and "with promising application prospects." (Reviewer `EWwE`),
- addressing a meaningful question "unexplored in the causal inference literature" and "derived a nice connection of causal classification with abstention with CvaR." (Reviewer `n12V`).

Meanwhile, Reviewer `EWwE` and `n12V` gave a slightly negative attitude towards the paper at an initial rating of 4 with **no further response**. Due to the updated policy of ICLR-26, we understand they cannot provide us with further discussions. For facilitate checking, we summarize their main concerns and our response details as follows:

## Reviewer EWwE

The main concerns from Reviewer `EWwE` and our responses are summarized as follows:

> Could the authors elaborate on these limitations of existing methods to provide a more comprehensive background?
- We have **added discussion** on limitations of existing methods in Section 1.
- We have extended the literature review to include a **broader comparison** with over 20 related works in Appedix B.

> Could the authors include such an overview of the model to improve the clarity and accessibility of the section?
- We have made clear overview of our model, including the **key components and rigorous definition** at the start of Section 2.
- We have **added a diagram** as Figure 1 to further clarify our framework.
- We have **added explanation** on the intuition of our method before Section 2.1.

> Could the authors address this aspect to inform readers of the computational efficiency of the approach?
- We have **added theoretical deduction** on both time and space complexity of our method in Section 3.4.
- In Appendix E.5, we also **added empirical analysis** on the running time and system memory of our method.

> Could the authors provide details on key hardware and software?
- We have **summarized all details** required by the author including computing environment, learning rate, batch size, etc. in Appendix E.2.

## Review n12V
While acknowledging the **soundedness and contribution** of this work, initial concerns from Reviewer `n12V` mainly focus on further clarification on technical details summarized as follows:

> Explanation on the hypothesis set (Q4) and the performance under simpler sets (Q9).
- We have added a clearer introduction on our framework through diagram in Figure 1 and extended discussions in Section 2, which explains why the abstention rule necessarily belongs to the hypothesis set.
- We **add experiments** on varying hypothesis set complexity in Appendix E.4 and demonstrate that our method has a superior performance under simpler hypothesis sets.

> Discussion on the strengths and examples of Assumption 4 and 5 (Q5).
- We have highlighted the strength and provided examples on Assumption 4 and 5 in the response. We also hightlight that we have **extended our method to accommodate possible violation on Assumption 5** in Section 3.2.

> The dependence of the error rate in Theorem 2 on problem-specific quantities (Q6).
- We have clarified that the error rate in Theorem 2 is linear to all related quantities through **rigorous derivation**. Extended discussions have been added beneath Theorem 2 in the revised manuscript.

> Further illustration on the convergence of the algorithm (Q7).
- We have **added discussion** on the convergence rate in rebuttal as well as empirical analysis on the convergence speed of our method in Table 8.

> Clarification on experimental details, including the realization of maximum function (Q8), MLP structure (Q10.1), and abstention strategy for baselines (Q10.2).
- We have explained the details on realization of maximum function and MLP structure, and **summarized all related details** in Section 4.1 and Appendix E.2.
- In Appendix E.6., we have **added experiments** showing that our method still outperforms baselines remarkably under **uncertainty-based abstention strategy**.

With the **novelty and soundedness** of our method acknowledged by Reviewer `n12V` and `EWwE`, we are confident that our responses can thoroughly address all concerns raised by the reviewers. Considering that the responded reviewer has a firmly positive attitude towards the paper (score 8) the reviewers with score 4 haven’t responded after the rebuttal was posted, we respectfully ask you to consider this context when making your final recommendation.

Many Thanks,

Authors of #7463

---

### Meta-Review · Area_Chair_oKKX · 2026-01-04

**Summary:**

The paper introduces a theoretically grounded solution to a problem of practical importance in some application domains such a heath care, that is treatment responder classification  with abstention on uncertain samples. The reviewers, while acknowledging the interesting approach, they also raised significant concerns about the clarity, assumptions and limitations of the proposed approach. The authors have significantly revised the manuscript to address many of them, however, as discussed below some concerns remain open.

My general impression on the paper is that the original version was not ready for publication, and the authors did major revision to address the reviewers' concerns. However, some aspects in my opinion remain still unclear and I believe that given the major nature of the revision the paper would need another round of reviews to confirm that it is above the acceptance bar.

**Reviewer Concerns:**

The authors clarify during the rebuttal important concerns regarding: i) the gap in the literature; ii) the time complexity of their approach; iii) sensitivity to hidden confounding; and iv) further clarifications on the theoretical assumptions. I believe however, that these changes are major and require a complete round of reviews, as unfortunately the discussion was cut short. Personally, I do not see how the new Figure 1 clarifies the training details of the classifier, as the sketch shows how it work in inference time. Similarly, the discussion about hidden con founding, while interesting would require careful review, as well as further empirical evaluation. The authors in the new section refer to the results, but it is unclear to me, which ones. Importantly, the empirical results require additional clarifications, especially the comparison between TRECA and TRECA+, since the later provides significantly worse results that TRECA  (by one order of magnitude according to Table 3) even when monotonicity assumptions do not hold. This makes me wonder, whether and when TRECA+ is ever preferred over TRECA.

In summary, I believe the paper is evolving in a positive direction but I believe it requires further revision and reviews to be ready for publication.

**Reviewer Scores:**

The positive reviewer acknowledge the changes and further improved their assessment. Unfortunately, the most negative reviewers did not reply to the authors prior to the closure of the discussion.  As a consequence, it is hard for me to predict what their opinion on the rebuttal would have been. In my opinion, some important concerns remain open and thus, I advocate for another full round of reviews.

---

### Decision · Program_Chairs · 2026-01-26

Reject